# NEURAL CONSTRAINT INFERENCE: INFERRING ENERGY CONSTRAINTS IN INTERACTING SYSTEMS

## ABSTRACT

Systems consisting of interacting agents are prevalent in the world, ranging from dynamical systems in physics to complex biological networks. To build systems which can interact robustly in the real world, it is thus important to be able to infer the precise interactions governing such systems. Existing approaches typically discover such interactions by explicitly modeling the feedforward dynamics of the trajectories. In this work, we propose Neural Constraint Inference (NCI) model as an alternative approach to discover such interactions: it discovers a set of relational constraints, represented as energy functions, which when optimized reconstruct the original trajectory. We illustrate how NCI can faithfully predict future trajectory dynamics, achieving more consistent long-rollouts than existing approaches. We show that the constraints discovered by NCI are disentangled and may be intermixed with constraints from other trajectories. Finally, we illustrate how those constraints enable the incorporation of external test-time constraints.

## 1 INTRODUCTION

Dynamical systems are ubiquitous in both nature and everyday life. Such systems emerge naturally in scientific settings such as chemical pathways and particle dynamics as well as everyday settings such as in sports teams or social events. Such dynamical systems may be decomposed as a set of different interacting components, where the interactions with respect to each other lead to complex dynamics. Modeling the underlying dynamics of such systems is hard: often times we only have access to example trajectories, without knowledge of the underlying interactions or the dynamics that govern them.

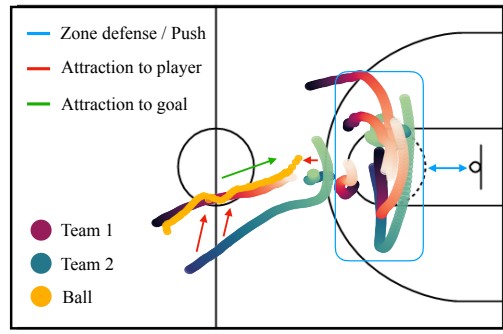

Figure 1: **Interactions between NBA players.** Complex dynamics, such as the player trajectories in the NBA, may be explained using a simple set of interactions. In this setting, one team of players aims to block a separate team from scoring.

Consider the scenario given in Figure 1, consisting of a set of NBA players playing a basketball game. While the motion of individual players may appear stochastic in nature, each player aims to score the basket on the opposite team's side of the court. Thus, we may utilize sets of interactions to explain their behaviors – a group of players on the defensive team serve as a zone defense, preventing players from the opposite team from getting close to the basket. Simultaneously, a group of offensive players moves towards the goal, while a group of defensive players moves to intercept them and prevent them from scoring. By applying our underlying knowledge of these interactions between players, we may forecast the future dynamics of the basketball game significantly more accurately.

Most works modeling such complex dynamics do not explicitly disentangle individual interactions between objects. Instead, they rely on a learned network to implicitly disentangle them (Battaglia et al., 2016; Gilmer et al., 2017; van Steenkiste et al., 2018). In contrast, Kipf et al. (2018) propose Neural Relation Inference (NRI), which learns a structured set of explicit interaction models between objects and show how such explicit interaction modeling enables more effective downstream predictions. In this work, we argue that we should instead model and disentangle interactions between objects

as a set of learned relational constraints, with dynamical prediction corresponding to a constraint satisfaction problem. To this end, we propose Neural Constraint Inference (NCI), where we encode each of these constraints as an energy function (Du et al., 2021).

To predict future dynamics with NCI, we then solve a constraint satisfaction problem, where we optimize for a trajectory prediction which minimizes our predicted energy. Prior work on implicit physical simulation has suggested that such implicit physics modeling (i.e. modeling dynamics through a constraint satisfaction problem) is significantly more accurate at simulating strong interactions in dynamics than explicit physics models (i.e. modeling dynamics as explicit feed-forward roll-outs) (Rubanova et al., 2022).

In different experiments, we illustrate how our constraint based decomposition of interactions provides unique benefits over prior learned approaches for decomposing dynamics. First, we illustrate how such a decomposition improves the temporal consistency, achieving significantly lower long-term temporal prediction error. We show that the decomposition is disentangled, enabling us to intermix interactions between separate trajectories together. We further show that constraints can linearly be decoded into underlying ground-truth interactions. Finally, we illustrate that such a decomposition enables us to add flexible test-time constraints to incorporate new changes in the environment.

In summary, in this work, we contribute the following: **(i).** We propose Neural Constraint Inference (NCI), which discovers, in an unsupervised manner, the underlying interactions between particles in a system as a set of energy constraints. **(ii).** We illustrate how such a constraint decomposition of interactions enables more accurate long-horizon trajectory prediction performance over prior methods. And **(iii).** we illustrate how such a constraint decompositions of interactions is disentangled and enables the recombination of constraints between separate trajectories, as well as the addition of novel test-time constraints.

## 2 LITERATURE

**Dynamics and Relational Inference** Several works in the past years have studied the problem of learning dynamics of a physical system from simulated trajectories with graph neural networks (GNNs) (Guttenberg et al., 2016; Gilmer et al., 2017; van Steenkiste et al., 2018; Lu et al., 2021; Li et al., 2018; Yang et al., 2022). As an extension of the foundational work of Battaglia et al. (2016), interaction networks, Kipf et al. (2018) proposes to infer an explicit interaction structure while simultaneously learning the dynamical model of the interacting systems in an unsupervised manner, by inferring edge classes with a classifier. Selecting models based on observed trajectories is also the base of Alet et al. (2019); Goyal et al. (2019); Graber & Schwing (2020). Graber & Schwing (2020) extends Kipf et al. (2018) to temporally dynamic edge constraints, which yields better results in real-world datasets. NCI differs from these approach as the generation procedure uses an optimization solver to satisfy a set of soft constraints. Recent work Rubanova et al. (2022) also explores combining graph networks with energy optimization. However, it lacks the modularity of NCI, and the ability to infer edge types from observation. Instead, a global energy function is learned for all nodes, by leveraging ground truth attributes such as mass. Thus, it has no mechanism to predict trajectories in the absence of those attributes nor when different types of relations are present.

**Energy-Based Models** Energy-based models have a long history in machine learning. Early work focuses on density modeling Hinton (2002); Du & Mordatch (2019); Nijkamp et al. (2020) by aiming to learn a function that assigns low energy values to data that belongs to the input distribution. To successfully sample data-points, EBMs have recently relied gradient-based Langevin dynamics Du & Mordatch (2019). Recent works have illustrated that such a gradient-based optimization procedure can enable the composition of energy functions representing different concepts Du et al. (2020) and successfully high-dimensional domains such as images Liu et al. (2021); Nie et al. (2021). Unsupervised discovery of composable energy functions on images was explored in Du et al. (2021). In this work, we extend ideas of unsupervised concept learning in EBMs to constraints and apply them to dynamical modelling and relational inference.

## 3 CONSTRAINTS AS ENERGY BASED MODELS

We will consider constraints as specifying a set $X$ of underlying trajectories $x \in \mathbb{R}^{T \times D}$ which have a underlying property we desire. In section Section 3.1, we discuss how we can represent constraints

on trajectories using an EBM. We further discuss how we may compose multiple constraints together as EBMs in Section 3.2.

## 3.1 ENERGY-BASED MODELS

**Definition.** An Energy-Based Model (EBM) is defined probabilistically using the Boltzmann distribution $p_\theta(\boldsymbol{x}) = \frac{\exp(-E_\theta(\boldsymbol{x}))}{Z(\theta)}$, with an underlying partition function $Z(\theta) = \int \exp(-E_\theta(\boldsymbol{x}))d\boldsymbol{x}$, where $\theta$ denotes the weights that parameterize the energy function $E_\theta$. We will represent a constraint as an EBM, defined using a neural network parameterized energy function $E_\theta(\boldsymbol{x}) : \mathbb{R}^D \to \mathbb{R}$ that maps each datapoint to a scalar value representing an energy. A constraint then corresponds to the set of datapoints in which the assigned energy is low. Thus, datapoints $\boldsymbol{x}$ satisfying our constraint have high likelihood, and all other datapoints have low likelihood. Constraint satisfaction then corresponds to sampling from the EBM distribution $p_\theta(\boldsymbol{x})$.

**Solving Constraints.** In our framework, solving a constraint corresponds to sampling from the EBM which defines it, and thus finding high-likelihood data points under $p_\theta(\boldsymbol{x})$. We follow existing works and utilize a gradient based MCMC procedure, Langevin Dynamics (Welling & Teh, 2011; Du & Mordatch, 2019) to sample from the EBM distribution. In particular, to solve a constraint, we initialize a trajectory $\boldsymbol{x}^0$ from uniform noise. We then run $M$ iterative steps following:

$$\tilde{\mathbf{x}}^m = \tilde{\mathbf{x}}^{m-1} - \frac{\lambda}{2}\nabla_{\mathbf{x}}E_\theta\left(\tilde{\mathbf{x}}^{m-1}\right) + \omega^m, \quad \omega^m \sim \mathcal{N}(0, \sigma), \tag{1}$$

where at each step we iteratively optimize the trajectory with respect to the energy function, using an underlying gradient step size of $\lambda$ and noise scale of $\sigma$. We include hyperparameter details for sampling in Section A.1 of the appendix, and heuristically set the noise scale of $\sigma = 0$.

## 3.2 COMPOSING CONSTRAINTS

Next, we discuss how we may compose different sets of constraints together, where each constraint is parameterized by a separate EBM $E_\theta^j(\boldsymbol{x})$. Our composition operator builds on existing works on composing EBMs representing concepts Du et al. (2021).

**Sampling Composed Constraints.** Given a set of separate constraints, we wish to solve for a set of trajectories $\boldsymbol{x}$ which jointly satisfy each of the constraints. In our EBM formulation, this corresponds to finding a trajectory $\boldsymbol{x}$ which is low energy under each of the specified energy functions $E_\theta^j(\boldsymbol{x})$.

Such a setting is equivalent to finding a trajectory $\boldsymbol{x}$ which has high likelihood under each EBM probability distribution $p_\theta^j(\boldsymbol{x})$. This corresponds to sampling from the distribution defined by the product of the individual EBM distributions,

$$\prod_j p_\theta^j(\mathbf{x}) \propto e^{-\sum_j E_\theta^j(\mathbf{x})} = e^{-E_\theta'(\mathbf{x})}, \tag{2}$$

which corresponds to a new EBM with energy function $E_\theta'(\mathbf{x})$ (an analogous approach can be applied to generate images subject to a set of concepts (Du et al., 2020)). Thus, we may sample from the composition of a set of constraints using a sampling procedure as Equation 1, using the new energy function $E_\theta'(\mathbf{x})$, defined as the sum of each individual energy function. Intuitively, this corresponds to a continuous optimization procedure on each energy function together.

In our setting, different energy functions $E_\theta^j(\boldsymbol{x})$ are constructed by conditioning an energy function on separate latent vectors. These latents are directly inferred unsupervised from input trajectories by training an encoder jointly with the energy function parameters.

## 4 NEURAL CONSTRAINT INFERENCE

Next, we discuss Neural Constraint Inference (NCI), our unsupervised approach to decompose a trajectory $\boldsymbol{x}(1...T)_i$, consisting of $N$ separate nodes at each timestep, into a set of separate EBM $E_\theta^j(\boldsymbol{x})$ constraints. Neural Constraint Inference (NCI) is composed by two steps: **(i)** an encoder for obtaining a set of energy constraints and **(ii)** a sampling process which optimizes for a predicted trajectory, given the inferred energy constraints. Energy functions in NCI are trained using autoencoding, similar to Du et al. (2021). We provide an illustration of our approach in Figure 2 and pseudocode in Algorithm 1.

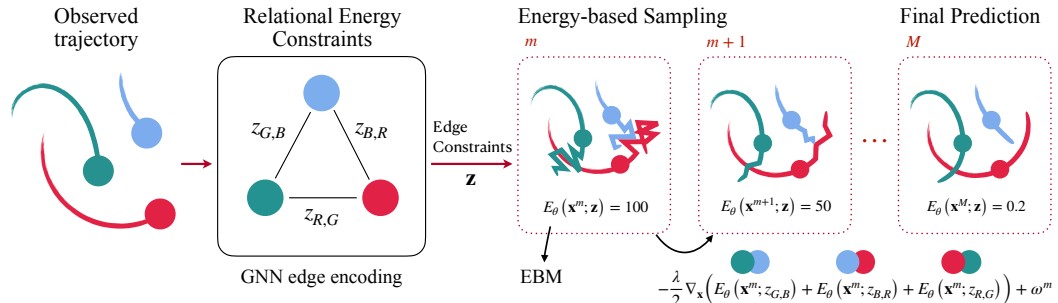

Figure 2: **Overview of NCI.** In the left, a portion $\mathbf{x}\,(1 \ldots T')$ of the input trajectory is observed by $\text{Enc}_\theta$ and encoded by a GNN into relational energy constraints, in the form of a set of latent vectors $\mathbf{z}$ for each edge in the graph. In the right, energy functions parametrized as GNNs for each edge latent vector in $\mathbf{z}$ are constructed. Energy functions are trained so that optimizing a trajectory $\mathbf{x}^0$ from uniform noise into a final trajecotry $\mathbf{x}^M$ reconstructs the future states of the observed trajectory. This refinement process uses Langevin Dynamics (Eq. 4). Given the full trajectory $\mathbf{x}^m$ at sampling step $m$, we update it by summing the gradient contributions of the energy function associated to each edge, resulting in $\mathbf{x}^{m+1}$.

## 4.1 RELATIONAL ENERGY CONSTRAINTS

To effectively parameterize different energy constraints for separate interactions, we learn a latent conditioned energy function $E_\theta(\boldsymbol{x}, \boldsymbol{z}) : \mathbb{R}^{T \times D} \times \mathbb{R}^{D_z} \to \mathbb{R}$. Then, inferring a set of different energy constraints corresponds to inferring a latent $\boldsymbol{z} \in \mathbb{R}^{D_z}$ that conditions an energy function.

Given a trajectory $\boldsymbol{x}(1...T)_i$, we infer a set of $L$ different latent vectors for each each pair of interacting nodes in a trajectory. Thus, given a set of $N$ different nodes, this corresponds to a set of $N(N-1)L$ different energy functions.

To generate a trajectory, we optimize the energy function $E(\mathbf{x}) = \sum_{ij,l} E_\theta^{ij,l}(\mathbf{x}; \mathbf{z}_{ij,l})$, across node indices $i$ and $j$ from 1 to $N$ and latent vectors $l$ from 1 to $L$. However, assigning one energy function to each latent code becomes prohibitively expensive as the number of nodes in a trajectory increases. Thus, to reduce this computational burden, we parameterize $L$ energy functions as shared message passing graph networks, grouping all edge contributions $ij$ in a single network. The energy is then computed as a summation over all individual node energies after message passing. To evaluate the energy corresponding to a single edge factor $\mathbf{z}_{ij,l}$ we mask out the contributions of all other edges to the final node energies. Architecture and further details can be found in Section A.2 of the appendix.

**Algorithm 1** Training algorithm for NCI.

**Input:** Full trajectories $\mathbf{x}$, Observed trajectories $\mathbf{x}(1...T')$, Initial conditions $\mathbf{x}(1...T_0)$, step size $\lambda$, number of gradient steps $M$, encoder $\text{Enc}_\theta$, energy functions $E_\theta^{ij,l}$, noise $\omega^m = 0$, true data distribution $p_D$
**while** not converged **do**
  $\mathbf{x}_i \sim p_D$
  ▷ *Encode components $\mathbf{z}_{ij,l}$ from $\mathbf{x}(1...T')$*
  $\{\mathbf{z}\} \leftarrow \text{Enc}_\theta(\mathbf{x}(1...T'))$
  ▷ *Optimize sample $\mathbf{x}_i^0$ via gradient descent:*
  $\mathbf{x}_i^0 \sim \mathcal{U}(0, 1)$
  **for** gradient step $n = 1$ to $N$ **do**
    $\tilde{\mathbf{x}}^m \leftarrow \tilde{\mathbf{x}}^{m-1} - \frac{\lambda}{2} \nabla_\mathbf{x} \sum_{ij,l} E_\theta^{ij,l}(\tilde{\mathbf{x}}^{m-1}; \mathbf{z}_{ij,l}) + \omega^m$
  **end for**
  ▷ *Optimize objective $\mathcal{L}_{MSE}$ wrt $\theta$:*
  $\Delta\theta \leftarrow \nabla_\theta \|\tilde{\mathbf{x}}^m - \mathbf{x}\|^2$
  Update $\theta$ based on $\Delta\theta$ using optimizer
**end while**

Figure 3: **Training Algorithm.** NCI is trained to infer a set of constraints, represented as energy functions, using a trajectory reconstruction objective. A set of latents $\{\boldsymbol{z}\}$ is inferred from the beginning of a trajectory $\mathbf{x}(1...T')$, and define different constraints. A trajectory is optimized w.r.t. to energy functions and supervised with the trajectory $\mathbf{x}$.

To condition to message passing shared graph network on each inferred latent $\boldsymbol{z}_{ij,l}$, each edge $e(i,j)$ in the graph is conditioned by the corresponding encoded edge latent code $\mathbf{z}_{ij,l}$, by means of FiLM modulation (Perez et al., 2018).

## 4.2 INFERRING ENERGY CONSTRAINTS

We utilize $\text{Enc}_\theta(\mathbf{x}) : \mathbb{R}^{T \times D} \to \mathbb{R}^{D_z}$ to encode the observed trajectories $\mathbf{x}$ into $L$ latent representations per edge in the observation. We utilize a fully connected GNN with message-passing to infer latents using the encoder module in Kipf et al. (2018). Instead of classifying edge types and using them as a gate ouputs, we utilize a continuous latent code $\mathbf{z}_{ij,l}$, allowing for higher flexibility.

### 4.3 TRAINING OBJECTIVE

To train NCI, we infer a set of different EBM constraints by auto-encoding the underlying trajectory. In particular, given a trajectory $\mathbf{x}(1...T)_i = (\mathbf{x}(1)_i, ..., \mathbf{x}(T)_i)$, we split the trajectory into initial conditions $\mathbf{x}(1...T_0)$, corresponding to the first $T_0$ states of the trajectory and $\mathbf{x}(T_0...T)$, corresponding to the subsequent states of the trajectory, where each state of the trajectory consists of $N$ different nodes. The edge constraints are encoded by observing a portion of the overall trajectory $\mathbf{x}(1...T')$, where $T' \leq T$.

We infer a set of different $L$ latents per edge of input observations utilizing the observed states $\mathbf{x}(1...T')$ using the encoder specified in Section 4.2, generating a set of latents $\{\mathbf{z}\}$. We then aim to train energy functions so that the following unnormalized distribution assigns low energy and high likelihood to the full trajectory $\mathbf{x}$:

$$p(\mathbf{x}|\{\mathbf{z}\}) \propto \prod_{i,j,l \forall i \neq j} p(\mathbf{x}|\mathbf{z}_{ij,l}) = \exp\left(-E_\theta^{ij,l}(\mathbf{x}; \mathrm{Enc}_\theta(\mathbf{x}(1...T'))_{ij,l})\right), \quad (3)$$

where $\mathbf{z}_{ij,l} = \mathrm{Enc}_\theta(\mathbf{x}(1...T'))_{ij,l}$ and $E_\theta^{ij,l}$ is the energy function linked to the $l_{\mathrm{th}}$ constraint of the encoded edge between nodes $i$ and $j$, respectively.

Since we wish to learn a set of constraints with high likelihood for the observed trajectory $\mathbf{x}$, as a tractable supervised manner to learn such a set of valid constraints, we directly supervise that sample using Equation 1 corresponds to the original trajectory $\mathbf{x}$, similar to Du et al. (2021). In particular, we sample $M$ steps of Langevin sampling starting from $\tilde{\mathbf{x}}^0$, which is initialized from uniform noise and the initial conditions fixed as the ground-truth $\boldsymbol{x}(1...T_0)$:

$$\tilde{\mathbf{x}}^m = \tilde{\mathbf{x}}^{m-1} - \frac{\lambda}{2}\nabla_\mathbf{x}\sum_{ij,l} E_\theta^{ij,l}(\tilde{\mathbf{x}}^{m-1}; \mathbf{z}_{ij,l}) + \omega^m, \quad \omega^m \sim \mathcal{N}(0, \lambda) \quad (4)$$

where $m$ is the $m_{\mathrm{th}}$ step and $\lambda$ is the step size. We then compute MSE objective with $\tilde{\mathbf{x}}^M$, which is the result of $M$ sampling iterations and the ground truth trajectory $\mathbf{x}$:

$$\mathcal{L}_{\mathrm{MSE}}(\theta) = \|\tilde{\mathbf{x}}^M - \mathbf{x}\|^2. \quad (5)$$

We optimize both $\tilde{\mathbf{x}}$ and the parameters $\theta$ with automatic differentiation. The overall training algorithm is provided in Algorithm 1.

## 5 EXPERIMENTS

In this section we firstly describe our datasets (Section 5.1) and baselines (Section 5.2). Following, we describe the quantitative results (Section 5.3). In Section 5.4, we show experiments on **(i.)** recombination, **(ii.)** edge classification and **(iii.)** contribution of the constraints. Next, in Section 5.5, we describe OOD sample detection. Finally, we show how to incorporate test-time constraints in Section 5.6. In the appendix we give implementation details (Section A.1), experimental details (Section A.3) and provide an ablation study (Section A.5) and additional examples (Section A.4).

### 5.1 DATASETS

We tested our model in three different domains. First, we carry on experiments in two simulated environments: **(i.)** Particles connected by springs, and **(ii.)** Particles with charges. Next, we test several properties of our model in **(iii.)** NBA SportVu motion dataset, which displays real motion from tracked basketball players along several NBA games. Finally, we test our performance in **(iv.)** JPL Horizons, a physics-based realistic dataset.

**Simulated data.** Following the experimental setting described in Kipf et al. (2018), we generate states (position and velocity) of a dynamical system for $N = 5$ particles for 70 time-steps. Our model observes the first 49 states, fixes one state and predicts the following 20. We generate 50k training samples and 10k for validation and test splits.

This setting is interesting because the rules by which particles interact are known and simple. However, they can generate very complex behaviour.

- **Springs**: The particles move inside a box with elastic collisions. They are connected by a spring with probability 0.5, and interact according to Hooke's law.
- **Charged**: The particles move inside a box as in Springs. They are assigned a positive or negative charge $q_i \in \{\pm q\}$ with probability 0.5 and interact via Coulomb forces.

| | Springs | | | Charged | | |
|---|---|---|---|---|---|---|
| Prediction steps | 1 | 10 | 20 | 1 | 10 | 20 |
| Static | 7.93e-5 | 7.59e-3 | 2.82e-2 | 5.09e-3 | 2.26e-2 | 5.42e-2 |
| IN | 1.32e-5 | 1.28e-3 | 4.71e-3 | **2.46e-4** | 1.06e-2 | 2.15e-2 |
| LSTM (single) | 2.27e-6 | 4.69e-4 | 4.90e-3 | 2.71e-3 | 7.05e-3 | 1.65e-2 |
| LSTM (joint) | 4.13e-8 | 2.19e-5 | 7.02e-4 | 1.68e-3 | 6.45e-3 | 1.49e-2 |
| Cond. GNN | 9.00e-6 | 6.13e-5 | 3.14e-4 | 3.67e-3 | 5.61e-3 | 1.05e-2 |
| NRI (full graph) | 1.66e-5 | 1.64e-3 | 6.31e-3 | 1.09e-3 | 3.78e-3 | 9.24e-3 |
| NRI (learned) | **3.12e-8** | 3.29e-6 | 2.13e-5 | 1.05e-3 | **3.21e-3** | 7.06e-3 |
| NCI (Ours) | 2.34e-7 | **1.57e-6** | **1.74e-5** | 8.85e-4 | **3.33e-3** | **6.54e-3** |

Table 1: Mean squared error (MSE) in predicting future states for simulations with 5 interacting objects. **NCI outperforms the baselines in mid to long-term prediction error**.

**NBA SportVU**  SportVU is an automated ID and tracking service that collects data of NBA players and the ball ($N = 11$) during a game. The dataset is generated by splitting each of the labeled events into 65 steps trajectories of coordinates x,y. We compute the velocities to generate the states. The dataset is composed of 50k samples for training and 1k samples for validation and test.

**JPL Horizons**  The JPL Horizons on-line ephemeris system provides access to solar system data. It characterizes the 3D location and velocity of solar system objects (targets) as a function of time, as seen from locations within the solar system (origins). This dataset consists on the trajectories captured between 1800 and 2022, with one datapoint every 10 days. We define the nodes as $N = 12$ targets of the solar system: 8 planets, 3 moons and the Sun. This data is captured from 13 origins: each one of the targets plus the solar system barycenter (SSB). We gather 1880 trajectories of 43 timesteps split as 1504/188/188 for train, validation and test.

## 5.2  BASELINES

We consider a **Static** baseline, which copies the previous state vector, **LSTM (single)**, an LSTM trained to predict the state vector difference at every timestep. We further compare with **LSTM (joint)** which differs from single in that it concatenates input representations from all objects after passing them through an MLP. For the synthetic experiments, where the edge types are known a priori, we also evaluate **NRI**, the architecture presented in Kipf et al. (2018) to infer the interaction graph (**learned**), and with a fully connected graph of a single edge type (**full graph**). We further add a GNN conditioned to the observed trajectories, and an interaction network (IN) Battaglia et al. (2016). For NBA SportsVU we will also evaluate on two social interaction-based methods: Social-LSTM (S-LSTM) Alahi et al. (2016) and Directional-LSTM (TrajNet++) Kothari et al. (2021).

## 5.3  QUANTITATIVE COMPARISON

For all datasets, we will observe a portion of the trajectory and predict 20 timesteps.

We first test our approach in Springs and Charged datasets. We evaluate the Mean-Squared Error (MSE) against Kipf et al. (2018), their chosen baselines, a generic Conditional-GNN and an IN. Our models observes 49 timesteps and fixes the $50_{th}$ as initial conditions for prediction. We can see in Table 1 that NCI achieves better long-term prediction in both datasets, and slightly worse short-term prediction error. This is likely caused by the nature of the generative process. The predictions are temporally one-shot, while they are refined by several optimization iterations.

Similarly, for NBA (Table 2 (left)) the model observes 40 timesteps and fixes the following 5 as initial conditions for prediction. NCI outperforms the baselines in terms of prediction error. The models designed for social interaction perform poorly in long-term prediction, while they have shown to excel in other tasks such as collision avoidance.

For the JPL Horizons dataset in Table 2 (right), NCI outperforms the baselines substantially. Models have access to 23 timesteps. NCI observes 20 timesteps and fixes 3 as initial conditions for prediction. JPL Horizons is a challenging dataset given the unknown masses of the bodies involved, as well as effects from unobserved smaller bodies nearby them that introduce noise to the trajectories.

## 5.4  DISENTANGLEMENT OF ENERGY CONSTRAINTS

NCI assigns an energy function to each one of the representations learned by the encoder. Those constrain the generation process by conditioning the features of their associated EBM. The optimization procedure, hence, aims to satisfy all constraints. We train our model to ensure that different

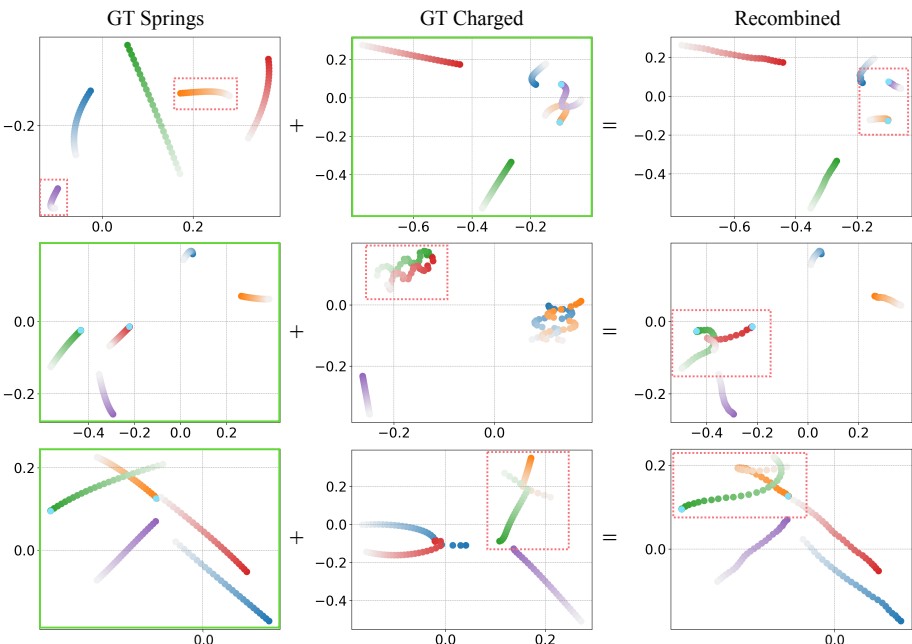

Figure 4: **NCI can recombine encoded energy constraints at test-time learned from different datasets.** Illustrated, samples from Springs (Col. 1) and Charged (Col. 2) and their recombinations (Col. 3). NCI encodes both trajectories. NCI is able to reconstruct trajectories framed in green with the initial conditions marked in blue, while swapping the edge constraints associated to the nodes in the red dashed box. Recombinations look semantically plausible and smooth.

| | NBA SportsVU | | | JPL Horizons | | |
|---|---|---|---|---|---|---|
| Prediction steps | 1 | 10 | 20 | 1 | 10 | 20 |
| S-LSTM | 6.60e-5 | 6.67e-3 | 2.57e-2 | - | - | - |
| TrajNet++ | 5.30e-5 | 5.88e-3 | 2.33e-2 | - | - | - |
| Static | 2.13e-4 | 3.04e-3 | 1.07e-2 | 3.33e-3 | 5.54e-2 | 9.05e-2 |
| LSTM (joint) | 8.07e-5 | 1.42e-3 | 5.31e-3 | 1.97e-6 | 3.98e-5 | 1.09e-4 |
| Cond. GNN | 1.71e-4 | 1.12e-3 | 3.11e-3 | 4.57e-6 | 4.66e-6 | 5.96e-6 |
| NRI | **3.56e-6** | 7.46e-4 | 2.74e-3 | **2.67e-7** | 7.35e-7 | 1.16e-6 |
| dNRI | **7.97e-6** | 1.07e-3 | 4.52e-3 | 1.35e-5 | 5.12e-5 | 1.64e-4 |
| NCI (Ours) | 1.27e-5 | **3.46e-4** | **1.86e-3** | 4.05e-7 | **4.70e-7** | **8.60e-7** |

Table 2: Mean squared error (MSE) in predicting future states for NBA dataset and JPL Horizons dataset, with 11 and 12 interacting objects respectively. NCI performs better than the baselines at short and long terms.

constraints contribute individually to the generative process by addition of their associated gradients. We argue that this procedure both **(i.)** aids discovery of disentangled edge representations and **(ii.)** allows composition among disjoint training distributions.

**Recombination**     To verify our claims, we show how NCI can compose energy constraints learned from two different distributions, at test-time. Figure 4 shows qualitative results of recombinations from Springs and Charged datasets. The process is as follows: we train two instances of our model ($NCI_S$, $NCI_C$) to reconstruct Springs and Charged trajectories respectively. Given sample trajectories drawn from each dataset (Col. 1 for Springs and 2 for Charged), we encode them into their relational energy constraints. For each row, we aim to reconstruct the trajectory framed in green while swapping one of the relational constraints (red dashed box) with an instance drawn from the other dataset. As an example, in the first row of the figure, we encode the Springs trajectory with $NCI_S$ and the Charged trajectory with $NCI_C$. Next, we fix the initial conditions corresponding to the Charged trajectory (blue dots) and sample by optimizing the relational energy functions. To achieve recombination, each model targets specific edges. We satisfy the constraints encoded by $NCI_S$ for the mutual edges corresponding to the nodes in red dashed boxes. The rest of edge constraints are encoded by $NCI_C$. The sampling process is done jointly by both models, each satisfying their corresponding edge constraints. The result is a natural combination of the two datasets, which constrain only the targeted edges. Reconstructed trajectories in Figure 4 (col. 3) are smooth and semantically reasonable.

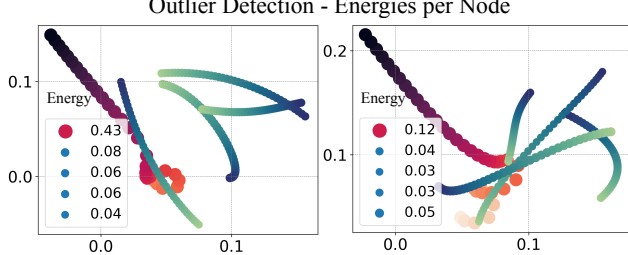

Figure 5: **We can use NCI for outlier detection.** A model trained with certain relation types can detect when a trajectory exhibits a new type of relation. The illustrated trajectories show the energy associated to each one of the nodes. Red trajectories: Charged particles, Blue-Green trajectories: Springs particles. We train NCI in Springs dataset. Our model assigns higher energies to those nodes that behave differently than the training set.

| Eval / Train | Springs |
|---|---|
| **Springs** | $0.00_{\pm 0.01}$ |
| **Charged** | $0.18_{\pm 0.30}$ |
| **S&C** (eval all) | $0.14_{\pm 0.30}$ |
| **S&C** (eval S) | $0.09_{\pm 0.18}$ |
| **S&C** (eval C) | $0.19_{\pm 0.37}$ |

Table 4: **Quantitative evaluation of out-of-distribution detection.** S&C rows correspond to the Springs and Charged mixed dataset. For this dataset we evaluate individually and jointly the Springs and Charged nodes.

**Edge classification**   To test the proper edge disentanglement capabilities of NCI, we propose to evaluate quantitatively the representation value of our inferred constraints. We measure their ability to classify correctly the edge type that they correspond to.

We train a linear layer to decode the constraints discovered by the encoder into edge types. The objective used is Binary Cross-Entropy w.r.t. the ground truth edges. Table 3 shows the accuracy obtained. In the table, Corr. (path) estimates the interactions by thresholding the matrix of correlations between trajectory features. In Corr. LSTM each trajectory is modeled individually and calculates correlations between output hidden states. For NRI, the accuracy is the result of their encoder's edge classification.

| Model | Springs | Charged |
|---|---|---|
| | 5 objects | |
| Corr. (path) | $52.4_{\pm 0.0}$ | $55.8_{\pm 0.0}$ |
| Corr. (LSTM) | $52.7_{\pm 0.9}$ | $54.2_{\pm 2.0}$ |
| NRI (sim.) | $\mathbf{99.8}_{\pm 0.0}$ | $59.6_{\pm 0.8}$ |
| NRI (learned) | $\mathbf{99.9}_{\pm 0.0}$ | $\mathbf{82.1}_{\pm 0.6}$ |
| NCI (linear dec.) | $97.4_{\pm 0.01}$ | $69.4_{\pm 1.0}$ |

Table 3: **Accuracy (in %) of true interaction recovery**. We train a linear decoder to predict edge types from constraints. Without supervision NCI learns meaningful representations that can often recover true interactions. Corr. indicates evaluation by means of the matrix of correlations between feature vectors.

We recover the true edge types with high accuracy in Springs and with moderate success for Charged dataset. For comparison, note that our evaluation setting for this experiment is considerably different from that of NRI. Unlike NRI, NCI's edge representations are discovered without the purpose of edge classification. We provide additional comparisons in Section A.4 of the appendix.

**Contribution of Energy Constraints**   Each of the disentangled constraints controls different aspects of the interactions. A qualitative example shown in Figure 6 depicts the gradient orientation of two energy constraints a model has been trained with. We can see how each constraint pushes the player trajectory into different directions, each one of them pointed to a different player of the rival team.

### 5.5   OUT-OF-DISTRIBUTION DETECTION

We further utilize the energy value at each constraint of NCI to detect out-of-distribution interactions in a trajectory. In our proposed architecture, energy is

Figure 6: **Energy constraints discovered by our approach control different aspects of the trajectories**. For a model trained with 2 energy functions, this illustration shows the gradients associated to each energy function applied to a ground-truth sample. Each constraint pushes the player of interest into one of the opponents.

evaluated at the node level. Therefore, if NCI has been trained with a specific dataset, the constraints associated to out-of-distribution type of edges are expected to have higher energy.

We design a new dataset (Charged-Springs) as a combination of Springs and Charged interaction types. In simulation, nodes are assigned both roles of Charged and Springs particles, but all the forces they receive correspond to one of the two types with probability $p = 0.5$. We train a model with the Springs dataset and evaluate the energies in the proposed mixed setting.

Figure 5 shows qualitatively how the energy is considerably higher for the nodes with Charged-type forces (drawn in red). Quantitative results are summarized in Table 4 for 1k test samples. We can

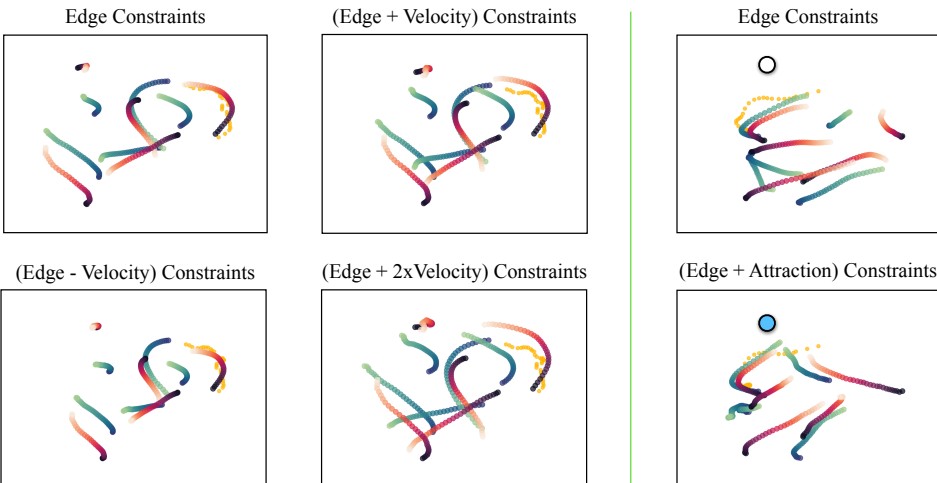

Figure 7: **NCI is able to incorporate new energy constraints in test-time**. We can see depicted reconstructions of NBA samples with added constraints. Left: (Col. 1, Row 1): Reconstruction of the encoded trajectory. (Col. 1, Row 2): Decrease of velocity. (Col. 2, Row 1): Low increase of velocity. (Col. 2, Row 2): High increase of velocity. Right: (Col. 3, Row 1): Reconstruction of the encoded trajectory, (Col. 3, Row 2): Attraction of the players to a goal point (blue dot). Painted orange, the ground-truth ball trajectory.

see that energies corresponding to Spring-type nodes are considerably lower than for Charged-type nodes, indicating that constraints are correctly capturing the behavior of the desired interactions.

## 5.6 FLEXIBLE GENERATION

Another advantage of our approach is that it can flexibly incorporate test-time user specified constraints. For this experiment, we investigate two different sets of constraints in Figure 7 when reconstructing a given 40 step trajectory of the NBA dataset.

**Velocity Constraints** We incorporate the following velocity energy constraint: $E = \epsilon\lambda \sum_{i,t} \sqrt{(\mathbf{v}_{x,i}^t)^2 + (\mathbf{v}_{y,i}^t)^2} = \epsilon\lambda \sum_{i,t} mod(\mathbf{v}_i^t)$, for particle $i$ in time $t$. The weight $\lambda = 1e-2/N$ scales the effect of this constraint over the rest and $\epsilon$ is a multiplicative constant that indicates the strength and direction of the constraint. Figure 7 (left), we show (**i.**) $\epsilon = 0$: Reconstruction (top-left); (**ii.**) $\epsilon = 4$: Decrease of velocity (bottom-left); (**iii.**) $\epsilon = -5$: Low increase of velocity (top-right); and (**iv.**) $\epsilon = -10$: High increase of velocity (bottom-right). Results satisfy test constraints.

**Goal Constraints** We also add at test-time an attraction energy constraint of the form: $E = \epsilon\lambda \sum_{i,t} \phi_i^t$. Here $\lambda = 1e-4/N$, $\phi$ is the angle between the velocity vector $\mathbf{v}_i^t$ of object $i$ at time-step $t$ and the orientation of the vector $\mathbf{p}_i^t \mathbf{q}$, where $\mathbf{p}_i^t$ is the location of object $i$ at time $t$ and $\mathbf{q}$ the point of attraction or goal. Figure 7 (right) illustrates the scenarios (**i.**) $\epsilon = 0$: Reconstruction (top); (**ii.**) $\epsilon = 3$: Attraction to the goal (bottom, goal in blue). The reconstructed trajectory follows the new constraint, while maintaining the constraints of the encoded trajectories.

## 6 DISCUSSION

**Conclusion.** In this work we introduced Neural Constraint Inference (NCI) which infers relational constraints specified as energy functions to model the dynamics of an interacting system. We illustrate how NCI can also faithfully predict future trajectory dynamics, achieving more accurate long-rollouts than the baselines. We further illustrate the disentanglement of the discovered constraints by intermixing them with constraints from other distributions of trajectories and show that constraints are interpretable. We also show that constraints obtained by NCI may discover out-of-distribution node interactions even within a scene. Finally, we illustrate the flexibility of modeling relational constraints – enabling the incorporation of external hand-crafted constraints at test-time.

**Limitations and Future Work.** Our current solution is currently limited to energy constraints associated to edges, and it would be beneficial to further explore its application to other type of constraints. Future work will focus on generalizing the use of the energy functions, for instance, targeting individual nodes in the graph.

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

# A APPENDIX

In this appendix, we present implementation details in Section A.1. Details about the architecture and training procedure can be found in Section A.2. Next, experiment settings are described in Section A.3. Following, additional image generation results in Section A.4, together with additional quantitative results. We further run an ablation study of each of our proposed components in Section A.5, together with an ablation study in test-time. Following, we evaluate the energy of NCI in longer time roll-outs in Section A.6. Finally we discuss the broader impact of our work in Section A.7.

## A.1 IMPLEMENTATION DETAILS

**Software:** We implemented this method using Ubuntu 18.04, Python 3.6, Pytorch 1.10, Cuda 11.2 and several additional libraries which will be provided as a environment requirements file.

**Hardware:** For each of our experiments we used 1 GPU RTX 2080 Ti (Blower Edition) with 12.8GB of memory. Models are trained for approximately 1 day.

## A.2 ARCHITECTURE AND TRAINING DETAILS

In this section we discuss in depth the architecture of the main modules of our method. We also discuss the idiosyncrasies of our training procedure.

**Architecture** The architecture of NCI is composed of 3 main modules: **(i.)** The encoder in Figure A3 is composed by convolutional and multi-layer perceptron blocks, with ELU activation functions. It encodes the observable trajectory $\mathbf{x}(1...T')$ into a set of $L$ latent codes per edge, with a total of $N \times (N-1)$ edges. **(ii.)** The short-term energy function in Figure A2 processes the trajectory in chunks of 5 time-steps. **(iii.)** The long-term energy function in Figure A1 processes the trajectory with several convolutional filters, while reducing its temporal resolution. It finally temporally pools the whole trajectory. It is designed to observe the overall shape of the trajectory. Both energy functions make use of the Swish activation function. The resulting energy is the summation of the short and long-term energies $E = E_{LT} + E_{ST}$. The terms node $\to$ edge and edge $\to$ node correspond to the different steps of message passing procedure. In node $\to$ edge, information from a connected node pair is concatenated in an edge representation. edge $\to$ node represents the summated contribution of all edge features connected to every node. The conditioning blocks modulate the energy function features by means of FiLM modulation Perez et al. (2018).

An illustration of the overall architecture can be seen in Figure A1.

**Constraint Splitting** To generate a trajectory, we optimize the energy function $E(\mathbf{x}) = \sum_{ij,l} E_\theta^{ij,l}(\mathbf{x}; \mathbf{z}_{ij,l})$, across node indices $i$ and $j$ from 1 to $N$ and latent vectors $l$ from 1 to $L$. Explicitly computing one energy function per edge becomes prohibitively expensive as the number of nodes in a trajectory increases. As introduced, the computational burden is reduced by utilizing a shared message passing graph network to compute a fixed set of features for all edges (Tables A1, A2, A3 and Figure A1). Hence, the energy corresponding to a single edge factor $\mathbf{z}_{ij,l}$ is obtained by masking out the contributions of all other edges. However, in order to recombine edge types across multiple datasets it is desirable to train the model to combine multiple energy function contributions. With this objective, in training time we randomly split the encoded edge constraints into two disjoint subsets. The generated trajectory is a product of joint optimization of two energy functions, each one conditioned to one of the subsets. Each energy function observes one edge constraint subset while masking out the contributions of the rest of edges.

**Regularization** To speed up training and regularize the energy values, we found useful to add the Contrastive Divergence loss $\mathcal{L}_{\text{CD}}$ Hinton (2002):

$$\mathcal{L}_{\text{CD}} = \mathbb{E}_{p_D(\mathbf{x})}\left[\sum_{ij,l} E_\theta^{ij,l}(\mathbf{x}; \mathbf{z}_{ij,l})\right] - \mathbb{E}_{\text{stop-grad}(q_\theta(\tilde{\mathbf{x}}))}\left[\sum_{ij,l} E_\theta^{ij,l}(\tilde{\mathbf{x}}; \mathbf{z}_{ij,l})\right]. \tag{6}$$

, where $p_D(\mathbf{x})$ is the true distribution of the data, and $q_\theta$ the distribution approximated by NCI. We also regularize the energy values by penalizing the squared energy resulting from above. These

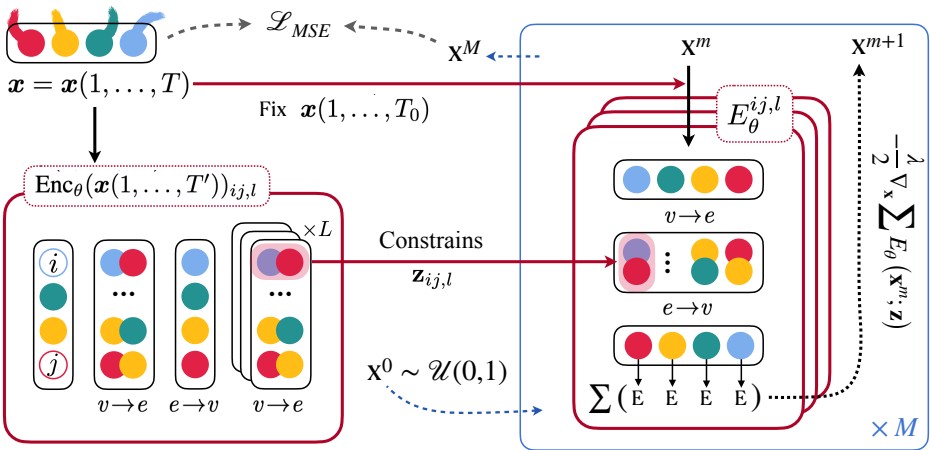

Figure A1: **Architecture and methodology of our approach.** In the left, $\text{Enc}_\theta$ observes a portion of the input trajectory $\mathbf{x}$ and encodes them into constraints, in the form of latent vectors $\mathbf{z}_{ij,l}$. In the right, a set of energy functions parametrized as GNNs are conditioned by $\mathbf{z}_{ij,l}$ at the edge level. We initialize a trajectory $\mathbf{x}^0$ as uniform noise and the ground-truth initial conditions $\mathbf{x}(1..T_0)$, and update it by minimizing the constrained energy functions. We sample by means of Langevin Dynamics. We supervise the reconstructed trajectories with an MSE objective with respect to the ground-truth trajectory.

regularizations are not necessary for the successful training of our model, however, they are helpful to stabilize training and therefore used for all experiments. For out-of-distribution detection experiments, regularization is remarkably useful. In-distribution samples are trained to have an energy close to 0, hence out-of-distribution samples are easily detected. Both regularizations are added to the primary objective (MSE) with a weight of $\lambda_{reg} = 1e-4$.

| Node → Edge |
| --- |
| 5x1 CNN Block Down (2) 64 |
| CNN Conditioning Block (2) 64 |
| CNN Conditioning Block (2) 64 |
| Temporal Avg. Pool |
| Edge → Node |
| MLP 64 |
| Dense → 1: $E_{LT}$ Long-Term Energy |

Table A1: Architecture for the **long-term energy function**. The energy computed evaluates the whole trajectory by leveraging 1D convolutional layers with a final temporal average pooling. Number of layers specified in parentheses.

| Node → Edge |
| --- |
| 5x1 CNN Block Down (2) 64 |
| Unfold Trajectory K:5, S:1 |
| Dense 64 |
| MLP Conditioning Block (2) 64 |
| MLP Conditioning Block (2) 64 |
| Edge → Node |
| MLP (2) 64 |
| Dense → 1: $E_{ST}$ Short-Term Energy |

Table A2: Architecture for the **short-term energy function**. The energy computed evaluates chunks of 5 steps of a trajectory, obtained with strides of size 1. Number of layers specified in parentheses.

| Node → Edge |
| --- |
| 5x1 CNN Block Down (3) 64 |
| Temporal Avg. Pool |
| MLP (2) 64 |
| Edge → Node |
| MLP (2) 64 |
| Node → Edge |
| MLP (2) 64 |
| Dense + LN → (L×Num. edges) |

Table A3: Architecture for the **encoder**. Number of layers specified in parentheses.

## A.3 EXPERIMENT DETAILS

In the following section we discuss the specific setting for each one of the experiments. In all cases, NCI uses Adam optimizer and a learning rate of $LR = 3e - 4$ with a scheduled decay of $\gamma = 0.5$ every 100k iterations ($\approx 80$ epochs). The illustrated trajectories shown in the figures have been plotted by accumulating the velocity dimensions of each predicted state to an initial ground-truth point.

**Baselines**  We choose the baselines described in Kipf et al. (2018), together with their main contribution, NRI. NRI is considered the de facto standard for relational inference, and LSTM the main baseline architecture for trajectory forecasting.

- **Static**: Copies previous state vectors.
- **LSTM (single)**: LSTM model trained to predict the state vector difference at every time-step. It consists of a tow-layer LSTM with shared parameters and 256 hidden units. The input to the model is passsed through a two-layer MLP with ReLU activations before it is passed to the LSTM. The last hidden vector of the LSTM for each time-steps is also passed through a two-layer MLP with ReLU activations, which outputs a predicted state difference. This is done individually for each particle. The LSTM has access to the ground-truth input states until prediction starts.
- **LSTM (joint)**: This model is similar to LSTM (single), with the difference that node states are concatenated before being processed by the model. This allows communication across node trajectories.
- **NRI (learned graph)**: For this model, an encoder infers interactions while simultaneously learning the dynamics from observational data. The encoder outputs a latent code that represents the underlying interaction graph and the reconstruction is based on graph neural networks.
- **NRI (full graph)**: This instanciation of NRI is similar to the one above, with the difference that the latent graph is fixed. The encoder is only allowed to output 1 type of edge representation.
- **dNRI**: Extension of NRI to a dynamic relation setting. The encoder infers separate relation graphs for every time-step. The reconstruction is based on graph neural networks, also in a dynamic fashion.
- **Conditional GNN**: For this model, we encode the edges similarly as in NCI, with the observed part of the trajectory. We decode them in one shot by means of a GNN with message passing.
- **IN**: This follows the official implementation provided by Battaglia et al. (2016). In this case, objects and relations are encoded separately, and propagated in time with a GNN. This model does not observe any part trajectory (unconditional).
- **Social LSTM and Directional LSTM (TrajNet++)**: This two models are implemented by Kothari et al. (2021) and executed following instructions in their github `https://github.com/vita-epfl/trajnetplusplusbaselines`. The architectures are LSTM-based, and use different types pooling functions to gather information surrounding each node. The code is addapted to a state with dimensionality 4 instead of 2.

**Quantitative Comparison**  We utilize the same setting and simulated datasets as detailed in Kipf et al. (2018), including their baselines: Static, LSTM (single), LSTM (joint) and variations of NRI. For the C-GNN, we train for 400 epochs with batch size of 40 and $D_z = 64$. We encode 50 time-steps and predict the following 20.

For Springs and Charged datasets, NCI is trained with 2 energy functions and latent size per edge constraint of $D_z = 64$. We encode 49 time-steps into a set of constraints, fixes 1 time-step from the ground-truth and predicts the following 20. Number of sampling steps $M$ varies from 3 to 6 along the first 300k iterations and a step-size of $\lambda = 0.4$. We use a batch size of 40 and train for 400 epochs.

Similarly, we run experiments in NBA SportsVU dataset with the same baseline setting for LSTM (joint) and C-GNN and the same data normalization scheme. C-GNN is trained similarly as before

with a batch size of 8 for 25 epochs. For prediction in NBA experiments, NCI encodes 40 time-steps into a set of constraints of dimension $D_z = 64$, fixes 1 time-step from the ground-truth and predicts the following 20. In training and testing, only the 20 predicted time-steps are generated and supervised with the ground-truth trajectory. Our model is trained for 25 epochs, with a batch size of 6 and single set of edge constraints. Number of sampling steps $M$ varies from 3 to 5 along the first 200k iterations and a step-size of $\lambda = 2$.

For this dataset, results in social interaction-based networks (S-LSTM, D-LSTM) are very poor in long term. This behavior has been discussed with the authors of D-LSTM (Kothari et al. (2021)), and concluded that it is expected when solving a task of prediction. This model is usually employed for tasks such as pedestrian collision avoidance with teacher forcing training. After switching the objective function to a purely MSE loss and changing the training strategy with the authors' help, the results improved very slightly. The model converged after 25 epochs.

NRI and dNRI are trained with a batch size of 40 and 32 respectively, a learning rate of $LR = 5e - 4$ and hidden sizes of 256. We train them for 35 epochs, until convergence which which takes approximately 24h. For both architectures, we use 2 edge types as recommended in the respective papers. Specifically, in NRI they try for higher number of edge types and conclude that the model is overfitting.

Finally, experiments in JPL Horizons dataset are with the same baseline setting as NBA SportsVU. In this case, we train LSTM (joint), C-GNN and NCI for 2000 epochs (given the small size of the dataset). In none of the cases we see signs of overfitting. C-GNN and NCI use a batch size of 10 and 6 respectively. NCI has a number of sampling steps $M$ that varies from 3 to 5 along the first 300k iterations and a step-size of $\lambda = 2$. In this experiment NRI and dNRI are trained for 2000 epochs with the same hyperparameters as in the NBA experiments. For both experiments we observe that dNRI performs worse than NRI. We tried to the best of our ability to obtain the expected results, but after hyperparameter tuning including modifications in the number of edge types, we conclude that the differences are negligible and therefore we keep the default settings.

**Recombination**  While not necessary, we found recombinations slightly more natural-looking by leveraging a variation of the original architecture. In this case, each EBM has a branch that evaluates trajectory energies unconditionally. That branch is solely used by edges that have been masked out (i.e. conditioned in another EBM). The unconditional architecture is therefore very similar to that in A1 and A2 but disregarding the conditioning blocks. We train NCI for 120 epochs both in Charged and Springs datasets separately, with latent size per edge constraint of of $D_z = 8$. For this experiment, NCI is trained to reconstruct 30 observed time-steps and predict the following 10, as we find prediction helpful for proper constraint learning. We use a dataset instance with double sampling frequency than for the other experiments. For data augmentation, we select randomly the initial point of the trajectories in the range $T_0 = [1, \ldots, 50]$, as velocity distributions diverge along the trajectories. The encoder observes the input data both rotated and instance-normalized. In test-time, we sample $M = 6$ times with a step-size of $\lambda = 16$.

**Out-of-distribution Detection**  We utilize the same Springs and Charged dataset variations as for the recombination experiments. For evaluation we also utilize the Charged-Springs dataset explained in the main body of the paper. The energy values are obtained at the node level by evaluating a ground-truth trajectory. The hyperparameters of the model are those of the recombination experiments.

We found especially useful to use the regularization in Equation 6. This maintains the in-distribution energy close to 0, while increasing energy from out-of-distribution nodes.

**Flexible Generation**  For this experiment, we train a model in NBA SportsVU dataset with the same hyperparameters as for quantitative comparison. However, in this case we only reconstruct 40 time-steps. The formulation of the hand-crafted constraints is described with detail in the main body of the paper. $\lambda$ corresponding to the weight of the new constrain is found by grid search by means of visual inspection. However, it is fixed for new instances of the experiment.

**Edge Classification**  For edge classification, we decode the constraints into edge types (1 or 0) with a dense layer. In all cases, we use models trained for prediction and decode the latent codes product of observing the 49 initial time-steps with the model. We train the dense layer with a BCE

| Model | Springs | Charged |
|---|---|---|
| | 5 objects | |
| LSTM (linear dec.) | 61.4 | 60.3 |
| NCI (linear dec.) | **97.4** | **69.4** |

Table A4: **Accuracy (in %) of true interaction recovery**. We train a linear decoder to predict edge types from latent codes. For LSTM, we linearly project the last hidden vector pairs associated to an edge between two nodes, excluding self-loops.

(binary cross-entropy) loss and $LR = 5e - 4$. For the LSTM in Table A4, we train a two-layer LSTM with shared parameters and 64 hidden units that models each trajectory individually. It receives ground-truth inputs until step 49 and is conditioned to the previous time steps. For this experiment, we keep the last hidden state of the second layer of the LSTM as our node representation. We generate 128 dimensional edges by concatenation of the node representations and decode them linearly into an edge type, training the layer for 75 epochs. Table A4 is discussed in Section A.4. Baseline setting for results in Table 3 are detailed in the main body of the paper.

GT Springs          GT Charged          Recombined

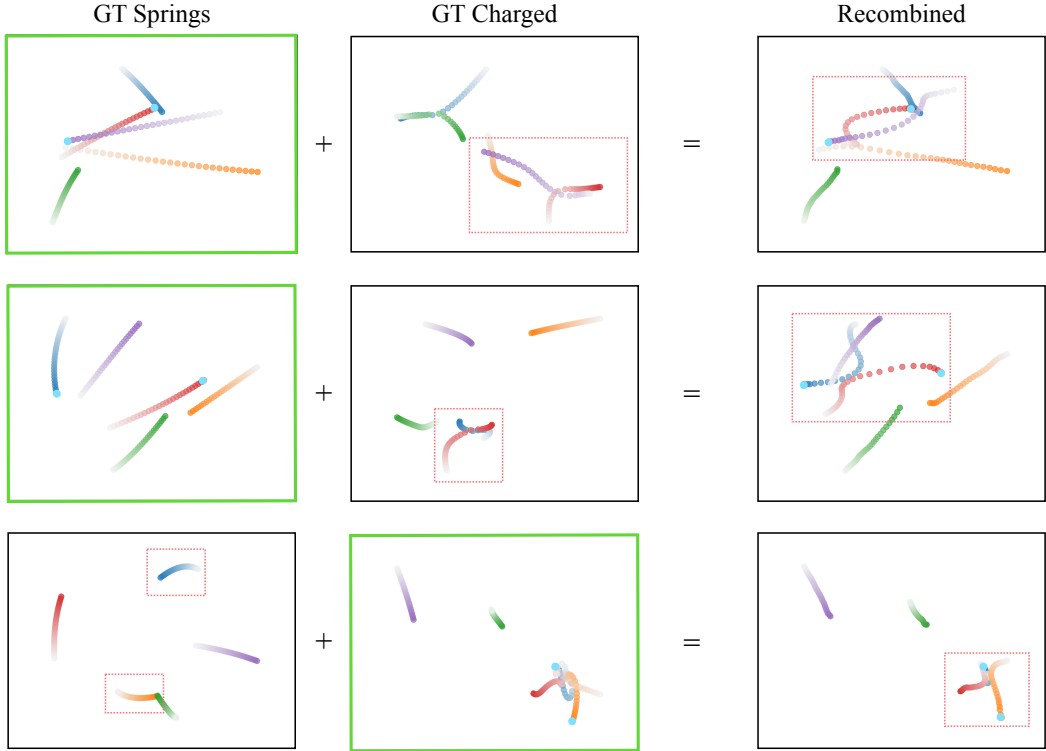

Figure A2: More examples of recombinations. Illustrated, samples from Springs (Col. 1) and Charged (Col. 2) and their recombinations (Col. 3). NCI encodes both trajectories. NCI is able to reconstruct trajectories framed in green with the initial conditions marked in blue, while swapping the edge constraints associated to the nodes in the red dashed box. Recombinations look semantically plausible and smooth.

## A.4   MORE EXAMPLES AND ADDITIONAL RESULTS

In this Section, we illustrate more examples of the main experiments and additional quantitative results.

**Qualitative Examples**   Figure A2 illustrates cross-dataset combinations of edge constraints. We encode ground-truth trajectories (columns 1 and 2) separately into their corresponding energy constraints. We train a different model for each data distribution. Next, we fix initial conditions $\mathbf{x}_0$ (blue points) of the scene framed in green and aim to reconstruct them. In test-time we generate a trajectory by minimizing simultaneously the energy functions corresponding to the two models. Each model targets specific edges. Particularly, one model adds constraints corresponding to the mutual edges of the particles highlighted by a red dashed box, while the other aims to reconstruct the rest of the trajectories.

Next, Figures A3 and A4 show qualitatively the predictions of NCI compared to the ground truth. In both cases, the initial 49 steps are the ground-truth trajectory. The black dot indicates the beginning of our model's predictions (the trajectory color gets lighter with time). Predictions are often accurate. In cases where there is a significant difference with the ground-truth (e.g. green node in the center of Figure A4), the predicted trajectory looks semantically plausible.

Figure A6 illustrates the ability of NCI to add hand-crafted constraints in test-time. In the top row, a trajectory reconstruction together with added goals that the nodes (players) are attracted to. In the bottom row, a trajectory reconstruction followed by the same trajectory with variations of the agent velocities. The resulting plots show how both the edge constraints and the newly added hand-crafted constraints are respected.

Springs: 50 GT + 20 predictions

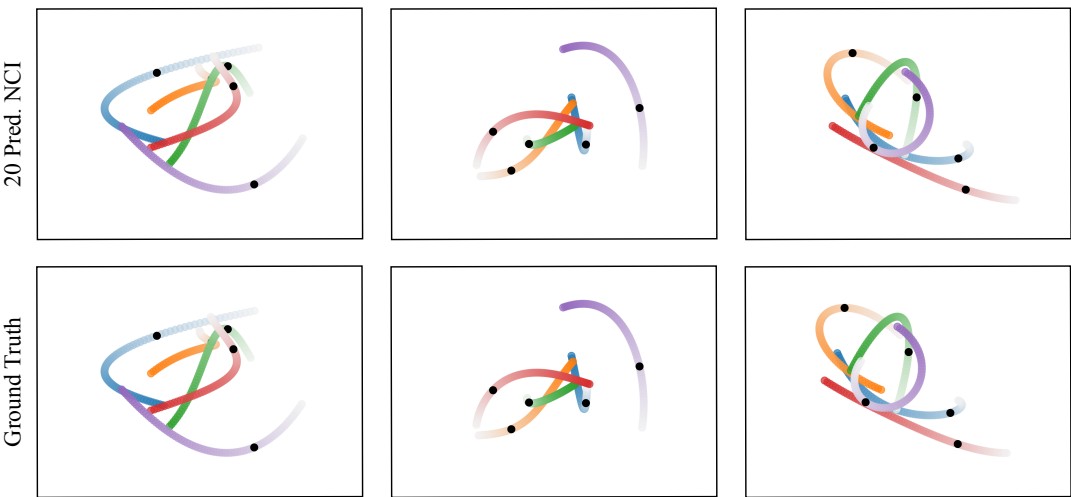

Figure A3: Qualitative example of 20 predictions in the future of NCI in the Springs particles dataset. In both rows, first 50 steps are the ground-truth. The black dot indicates the beginning of predictions for row 1. In all cases, predictions look almost identical to the ground-truth.

Charged: 50 GT + 20 predictions

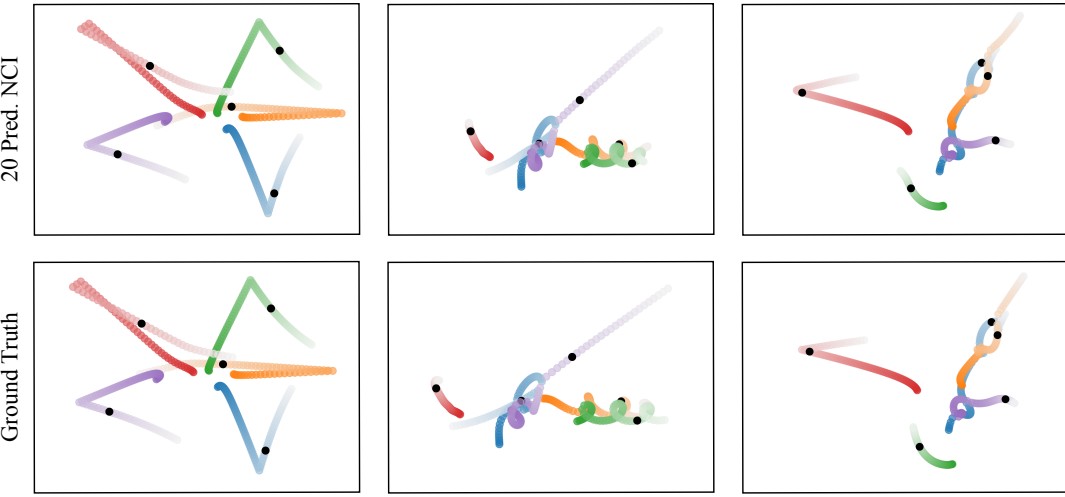

Figure A4: Qualitative example of 20 predictions in the future of NCI in the Charged particles dataset. In both rows, first 50 steps are the ground-truth. The black dot indicates the beginning of predictions for row 1. In all cases, predictions are fairly close to the ground-truth. In the cases where they differ (green node - center) the predictions are smooth and look reasonable.

Finally, Figure A5 illustrates an example of the sampling procedure. By leveraging Langevin Dynamics sampling we refine our predictions iteratively in a gradient-based optimization procedure. The model quickly understands the general shape of the trajectories and refines them locally using the final sampling steps.

**Edge Linear Decoding Results**   We show in Table A4 the quantitative comparison of NCI with respect to the LSTM (single) baseline for linear edge type decoding. Note that NRI learned edge representations are already a classification of explicit edge types and therefore do not allow for linear decoding. The resulting accuracy of NCI surpasses that of the LSTM by a significant margin.

Sampling Procedure in Trajectory Reconstruction

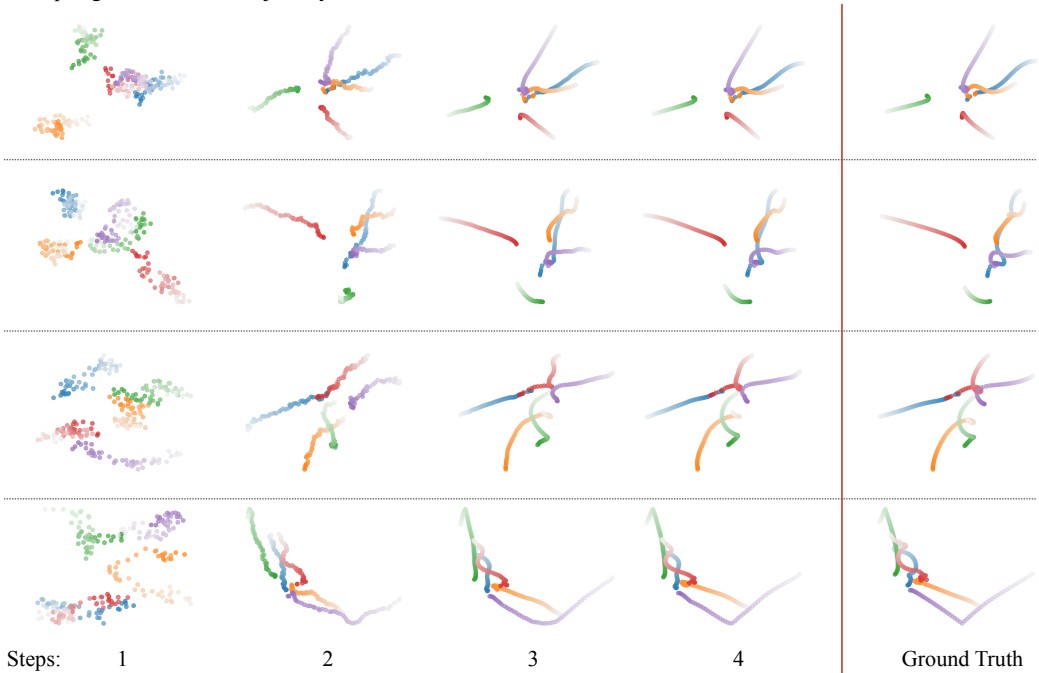

| Steps: | 1 | 2 | 3 | 4 | Ground Truth |

Figure A5: Examples of trajectory reconstruction procedure for 50 time-steps of the Charged dataset. We initialize the velocity as uniform noise and sample 4 times (in this case) using Langevin Dynamics. We obtain a faithful reconstruction of the ground-truth trajectories.

## A.5 ABLATION STUDY

Following, we add an ablation study carried on in the Charged dataset, with 370k iterations and a batch size of 64. Here, we analyse NCI's performance through a variety of design choices. Those are the following:

- Latent size: We explore different sizes for the edge constraints. The choices are LS $\in \{16, 32, 64, 128\}$.
- Langevin step size. The choices are $\lambda \in \{0.1, 0.2, 2.0, 6.0, 10.0, 14.0\}$.
- Objective: We evaluate the impact of adding the contrastive divergence objective to the MSE reconstruction loss.
- Edge masking strategy: In a setting with 2 EBMs, we evaluate 1) a random masking strategy, 2) a masking strategy based on the edge contribution to a specific node and 3) no masking strategy.
- Decoder baseline: We substitute the iterative energy minimization by a feed-forward graph decoder. This is equivalent to the Conditional GNN baseline.

For this experiment, we follow the setting in the quantitative comparison and simply modify the variable of interest. Tha analysis shows how a small langevin step is desirable in terms of performance ($\lambda = 0.2$). The model seems to be robust to the latent size choice, although there is a slight preference for $LS = 32$. When it comes to the objective, we show quantitatively that regularizing training with a contrastive divergence term improves performance. We can also observe how a masking strategy is better than none. Finally, we show by comparing to a decoder baseline how our sampling strategy is crucial for competitive results. Our ablation study is summarized in Table A5.

Similarly, we perform an ablation at test-time, with our best trained model for the Charge dataset. We analyse NCI's performance under the following variations:

- Number of Langevin steps: We train our model with $M = 6$ langevin steps and test it with $M \in [1, 2, 3, 4, 5, 6, 7, 8, 9, 10, 20]$.
- Number of nodes: We train NCI with 5 nodes and test it in datasets with $N = 3$ and $N = 7$

Results are summarized in Table A6. "Train" indicates the setting used in training.

## A.6 LONGER-TIME ROLL-OUTS

We evaluate our model against NRI in longer term prediction. Given the chaotic nature of trajectories, we evaluate the energy among consecutive states (squared pairwise difference) for 100 timesteps. This time, we employ our model auto-regressively in the Charged dataset, while it has been trained to predict 20 steps in one-shot. R1 indicates that we iteratively predict 1 time-step. Results in Table A7 show how NCI preserves approximately more than 3 times of the energy than NRI as time unfolds.

## A.7 BROADER IMPACT

Understanding interactions across agents in a trajectory is fundamental to explain their present behavior and predict their future. The importance of such understanding is higher when we do not have access to the true interaction types or they are simply not a discrete set. In those cases, being able to learn representations of interactions from observational data provides a window into the physics of the world we live in. These property is desirable in AI for applications such as molecular dynamics modeling or autonomous vehicles, which have a huge impact on our lives. Despite the fact that there is a wide range of approaches for inferring interactions and predicting trajectories, there is relatively little work on inferring these in a interpretable manner. Our model aims to learn these interaction constraints that allow for a higher degree of manipulation over the learned representations. This interpretability and manipulability properties are important to AI, but might raise concerns of abuse. Our approach, similar to many other approaches, may capture the implicit biases present in data. There is also the potential threat of attacks to systems that rely on interpretable models, which can be more easily targeted than those which are opaque.

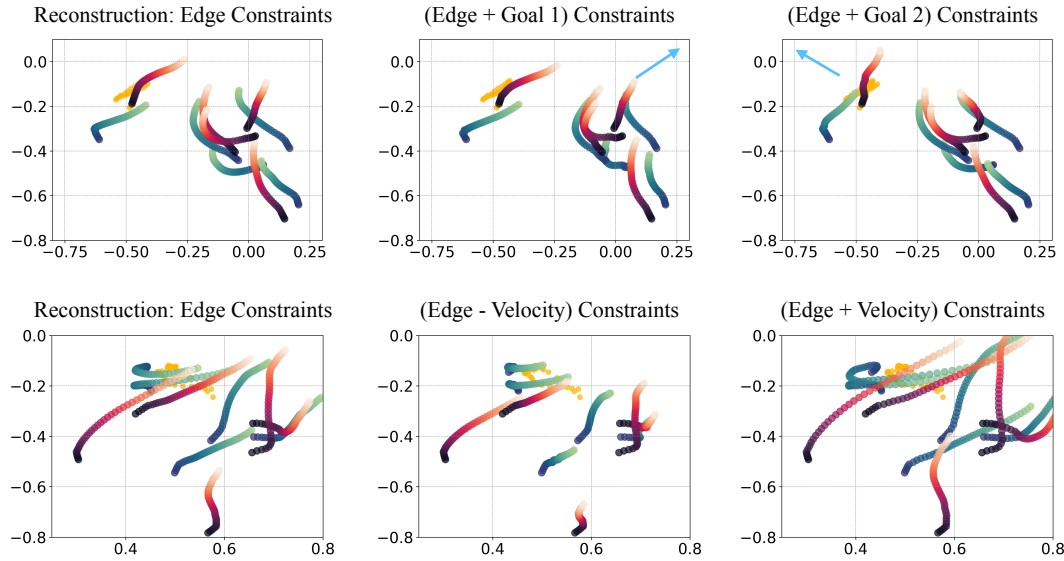

Figure A6: More examples of NBA reconstructed trajectories with added constraints in test-time. NCI can generate realistic trajectories by respecting both **(i).** The learned edge constraints + **(ii).** new hand-crafted constraints added in test-time. Illustrated (top row) we see the reconstruction of the trajectory with an added goal. We can see that the trajectories have the tendency to be attracted in the direction indicated by the blue arrow. The actual goal is located outside the frame. We can see (bottom row) constraints of higher and lower velocity than the reconstruction. In all cases, the trajectories follow the new constraints without loosing their original constraints.

| Ablation Study | 1 step | 10 step | 20 step |
|---|---|---|---|
| Latent Size | | | |
| 16 | 1.20e-3 | 3.75e-3 | 7.06e-3 |
| 32 | 9.87e-4 | 3.60e-3 | 6.82e-3 |
| 64 | 1.15e-3 | 3.82e-3 | 7.00e-3 |
| 128 | 1.15e-3 | 3.91e-3 | 7.24e-3 |
| Sampling step size | | | |
| 0.1 | 9.54e-4 | 3.53e-3 | 6.81e-3 |
| 0.2 | 9.07e-4 | 3.36e-3 | 6.62e-3 |
| 2.0 | 1.16e-3 | 3.82e-3 | 7.09e-3 |
| 6.0 | 1.15e-3 | 3.82e-3 | 7.00e-3 |
| 10.0 | 1.14e-3 | 3.78e-3 | 7.07e-3 |
| 14.0 | 1.19e-3 | 3.91e-3 | 7.29e-3 |
| Masking type | | | |
| Mask random | 1.15e-3 | 3.82e-3 | 7.02e-3 |
| Mask by node | 1.15e-3 | 3.78e-3 | 7.03e-3 |
| No masking | 1.17e-3 | 3.82e-3 | 7.09e-3 |
| Objective | | | |
| MSE+CD | 1.15e-3 | 3.82e-3 | 7.02e-3 |
| MSE (no CD) | 1.36e-3 | 4.02e-3 | 7.47e-3 |
| Decoder (no sampling) | 3.27e-3 | 5.31e-3 | 9.65e-3 |

Table A5: **Ablation study** investigating effects of different factors like latent size, sampling step size, masking type and objective in results.

| Ablation Study in Test | 1 step | 10 step | 20 step |
|---|---|---|---|
| Number of Langevin steps $M$ | | | |
| 1 | 1.06e-1 | 1.18e-1 | 2.47e-1 |
| 2 | 3.23e-2 | 4.34e-2 | 8.22e-2 |
| 3 | 7.00e-3 | 1.16e-2 | 2.68e-2 |
| 4 | 1.86e-3 | 4.63e-3 | 1.00e-2 |
| 5 | 1.03e-3 | 3.51e-3 | 7.05e-3 |
| 6 (Train) | **8.85e-4** | **3.33e-3** | 6.54e-3 |
| 7 | 9.00e-4 | **3.33e-3** | **6.45e-3** |
| 8 | 8.93e-4 | 3.36e-3 | 6.47e-3 |
| 9 | 9.00e-4 | 3.39e-3 | 6.60e-3 |
| 10 | 9.10e-4 | 3.44e-3 | 6.66e-3 |
| 20 | 1.03e-3 | 3.64e-3 | 7.04e-3 |
| Number of nodes $N$ | | | |
| 3 | 1.79e-3 | 3.71e-3 | 6.74e-3 |
| 5 (Train) | **8.85e-4** | **3.33e-3** | **6.54e-3** |
| 7 | 1.34e-3 | 4.66e-3 | 1.01e-2 |

Table A6: **Ablation study** investigating effects of different factors in test-time: Number of langevin steps and number of nodes.

| Energy Evaluation (1e-3 | 1-2 | 10-11 | 20 -21 | 30-31 | 40-41 | 50-51 | 60-61 | 70-71 | 80-81 | 90-91 | 100-101 |
|---|---|---|---|---|---|---|---|---|---|---|---|
| Baseline | 5.0 | 4.9 | 5.0 | 5.0 | 4.9 | 4.9 | 5.0 | 4.9 | 5.0 | 5.0 | 4.9 |
| NCI (R1) | 3.2 | 2.4 | 2.2 | 2.2 | 2.1 | 2.1 | 2.0 | 1.9 | 1.9 | 2.0 | 2.0 |
| NRI | 1.2 | 0.75 | 0.65 | 0.60 | 0.68 | 0.62 | 0.61 | 0.70 | 0.59 | 0.57 | 0.65 |

Table A7: **Energy evaluation NCI preserves more energy in long rollouts than NRI.**

