# OpenReview forum: "Neural Constraint Inference: Inferring Energy Constraints in Interacting Systems"
_ICLR.cc/2023/Conference — Submitted to ICLR 2023_

### Official Review · Reviewer_1k6S · 2022-10-20

**Confidence:** 4
**Correctness:** 2
**Technical Novelty And Significance:** 3
**Empirical Novelty And Significance:** 3
**Recommendation:** 5

**Clarity, Quality, Novelty And Reproducibility:**

**CLARITY**

The paper is well written.

**QUALITY**

Overall, the technical part of the paper is reasonable and complete. The experimental evaluation can be strengthened by adding additional comparisons with NRI (as mentioned above).

**NOVELTY**

The idea of solving the considered problem by using an energy-based model is new and original. In particular, the work contributes to extend previous results on the composable properties of energy-based models (which have been previously observed for unconditional generative tasks) to the relational dynamic setting.

**Strength And Weaknesses:**

**Strengths**
- The paper is well-written. I enjoyed reading it.
- The idea of learning an energy-based model in the dynamics setting is to my knowledge new. Maybe you can also consider some recent related work [1].
- The proposed solution enables to tackle new challenges for the toy cases (OOD and conditional generation with user-defined constraints) and therefore has a good potential.

**Weaknesses**
- Several experimental comparisons against NCI are missing and should be included (see below for further details).
- Code is not available.

Regarding the first weakness:
- Experiments (general) Computationally speaking, it is not clear how the proposed model compares against NRI. Specifically, while NRI uses an autoregressive model for the decoding process, NCI uses a gradient-based procedure for sampling, which requires the computation of second-order derivatives during training. How are these derivatives handled during training? And can you provide a discussion and perhaps a quantitative comparison both for training and inference with NRI?
- Experiments (general). The proposed model is in essence a conditional generative model. How does the model generalise to predictions at future time steps larger than the ones seen during training? As far as I understand, the model can be iteratively applied by setting the initial state of the sampling process to the last generated portion of the trajectory, thus enabling to see further in the future. Is this correct? If so, why not comparing on this task against NRI?
- Experiments (general). How does the model generalise to a different number of objects seen during training?
- Experiments (toy) on traditional generation and edge classification tasks. In Table 1, NCI seems to achieve similar performance to NRI. In which sense is the model outperforming NRI, can you elaborate more on that? Additionally, I found a bit confusing the experiments on edge classification, as NCI is clearly inferior to NRI. What is the message here?
- Experiments (real) why there is no comparison with NRI?

Regarding the second weakness:

Are you planning to integrally release the code?

Some minor comments/curiosities:
- The maximization of the conditional log-likelihood requires to minimize the energy score for training data while maximising it for generated data. Why this second optimisation term is not used in all the experiments, but only for OOD detection (as mentioned in the Appendix A)?
- In Table 1, please add the lower bound baseline with known graph.
- Can you think of other real cases other than the NBA dataset where additional constraints can be specified?
- Last paragraph in the introduction: decompositionS of interactions is disentagleMENT -> decomposition of interactions is disentangled.

[1] ModLaNets: Learning Generalisable Dynamics via Modularity and Physical Inductive Bias (ICML 2022)

**Summary Of The Paper:**

The paper proposes to learn an energy-based conditional generative model from trajectories of multiple particles. Specifically, the model (i) uses a graph neural network to encode partial trajectories into typed relations (each type is equivalently seen as a constraint) and then (ii) uses such information to define an energy score function by summing all the energy contributions of each relation. A standard gradient-based procedure for energy-based models is used to sample the full trajectory given the extracted relations, thus acting as a decoding process. The proposed model is learnt by maximising the conditional log-likelihood.

From a conceptual point of view, the work differs from related approaches, including the neural relational inference (NRI), in the sense that it tackles the same problem of NRI using an energy-based formulation instead of a VAE one. From a computational point of view, the work replaces the explicit decoder in NRI with a sampling procedure (involving the use of second-order derivatives). From a task point of view, the energy-based formulation enables to tackle some challenges currently out of reach for NRI, including the OOD detection and the conditional generation of additional user-defined constraints (e.g. constraints defined as differentiable functions on the trajectories) introduced at test time.

Experiments are conducted on 2 toy datasets, namely springs and charged particles, and 2 real datasets, including NBA SportVU and JPL Horizons.

**Summary Of The Review:**

------- POST REBUTTAL -------------

The answers in the rebuttal partially address my concerns. Doubts still remain on the nature of the method and the experimental methodology (both in terms of predictive performance and computational cost).

The proposed model learns a conditional distribution $p(x|x_{partial})$, where $x$ and $x_{partial}$ are the full and partial trajectories, respectively. The definition in Section 4.3 Eq. (3) is not appropriate. The model is not conditioned on the constraints, which are learnt and not given, but it is conditioned on the partial trajectories. This is not only a minor technical issue, but it has also an important practical consequence. In fact, the model is, in all effect, autoregressive and the experiments should take into account this fact.
My original concern about the comparisons with neural relational inference (NRI) still remain after the rebuttal. In both old and new experiments, NRI outperforms the proposed method in terms of predictive performance for step 1 (both on toy and real data, see for example "General comments 2/2 part. 2.2"). However, when considering larger future steps predictions, namely 10 and 20 (for real), the phenomenon is inverted. This sheds some lights on the unfair methodology used in the comparisons. The proposed method has been trained to deal with predictions (up to 20 steps) contrarily to NRI, which deal with them in an autoregressive way. Therefore, the authors should consider the same conditions for comparisons and treat their model as an autoregressive one.

Additionally, the analysis in terms of computation for the proposed approach provided in the rebuttal is not informative. The authors have shown an empirical comparison against NRI by considering fixed values of batch size and trajectory length. What does it happen when increasing the number of particles, their feature dimensionality or the trajectory length? I expect that the computation of second-order derivatives (required by the training/decoding component) becomes quickly intractable in these cases, whereas NRI is still able to handle them. However, such analysis is still elusive and missing.

Overall, I think that the idea of using an energy-based model is nice as enabling to handle new tasks, including OOD detection and dealing with new constraints at test time. However, I feel that the experimental analysis is still not complete and the main claims are not well supported (see e.g. in the abstract "achieving more consistent long-rollouts than existing
approaches"). Consequently, I decrease my score.

---

> ### Author Response · Authors · 2022-11-17
> **Comment for reviewer 1k6S**
>
> We appreciate the positive and constructive comments from the reviewer. We will modify our paper according to your comments and address concerns below.
>
> * [**Code**] Code base including training files and models to reproduce the main experiments will be released with the final version of the paper.
>
> > Computationally speaking, it is not clear how the proposed model compares against NRI.
>
> Our model relies on inner-loop optimization both in training and inference time. We leverage Pytorch's AutoGrad automatic differentiation engine to compute both the weight updates and the langevin steps. We will thus have to compute gradients of gradients (second order optimization as pointed out), which is also handeled automatically by AutoGrad for certain types of computational graphs. As suggested by recent work on EBM training, we only backpropagate through the last langevin step. In our training we use a curriculum. We increase the number of steps typically from 3 to 6. We use the latent codes to condition the EBM's edge features and sample jointly for all EBMs.
> Instead, NRI classifies the observed trajectory into an edge-type, and uses that class as a gate/selector to choose from a set of decoder functions. Those can be either an RNN or an MLP. The decoders are applied auto-regressively both in training and testing.
> To quantitatively compare the computational burden of each model we fix the computational budget and evaluate the time per iteration and sample in seconds. We use a batch size of 40 and predict 20 timesteps, averaged over 400 samples.
>
> | Time (s) per iteration  and sample | Train   | Test    |
> |-------------------------------------------|---------|---------|
> | NRI (CNN+MLP)  | 1.96e-3 | 7.06e-4 |
> | NRI (CNN+RNN) | 4.96e-2 | 4.33e-2 |
> | NCI (3 steps)  | 3.91e-3 | 1.78e-3 |
> | NCI (6 steps) | 5.91e-3 | 3.40e-3 |
> > How does the model generalise to predictions at future time steps larger than the ones seen during training?
>
> We can indeed predict auto-regressively, although NCI does not supervise the unrolling process during training. As suggested by reviewer C8J2, we can compare them in terms of energy preservation in the longer term, given the chaotic nature of our datasets. We provide this evaluation in the general comments for 100 time-steps.
> > How does the model generalise to a different number of objects seen during training?
>
> We provide this result in the ablation study of the general comments.
> > Experiments (toy) on traditional generation and edge classification tasks. In Table 1, NCI seems to achieve similar performance to NRI.
>
> **Our model outperforms NRI by displaying several useful properties while preserving equivalent predictive power**. Our view is that what NCI brings to the community are new properties in trajectory modelling and manipulation that emerge naturally from the method, as the qualitative experiments show: Recombination, OOD detection, test-time added constraints and interpretability of the energy gradients.
> The results in prediction are meant to illustrate that the model is also able to correctly forecast the trajectories, in par with dedicated models that are designed for forecasting.
>
> > I found a bit confusing the experiments on edge classification, as NCI is clearly inferior to NRI. What is the message here?
>
> It is unclear how to compare our method to NRI in terms of edge-type decoding where we do know the ground-truth edges. The reason is that NRI relies on edge classification as part of their pipeline. Instead, we linearly decode the learned continuous edge features into discrete edge types, a posteriori and with a single linear layer. As the comparison is not exactly fair, the take-away message of the table is that NCI is learning non-trivial edge features, that can often recover the true edge types with a simple logistic regression model. Additionally, in Table A4 (Section A.3 of the Appendix) we compare NCI to a jointLSTM with the same linear decoding scheme.
>
> > Experiments (real), comparison with NRI?
>
> We provide comparisons to NRI for the real settings in the general comments.
>
> > Why this second optimization term is not used in all the experiments?
>
> **We used it for all our experiments**. We just highlight that it is especially useful in OOD, because it helps us to understand in-liers as energies close to 0, instead of energies with potentially arbitrarily high absolute value. We modify that section to clarify it.
> > In Table 1, please add the lower bound baseline with known graph.
>
> Could you clarify? We did not completely understand the request, but are happy to add a comparison in the final version of the paper.
> > Can you think of other real cases other than the NBA dataset where additional constraints can be specified?
>
> For our current setting, one interesting test-time constraint could be an energy preservation constraint, where we force our predictions to preserve certain energy by penalizing trajectories with vanishing/exploding energy.

---

> ### Author Response · Authors · 2022-11-20
> **Clarifications for Post-Rebuttal Response**
>
> Hi Reviewer 1k6S, thank you for responding to our rebuttal. We think there might be some misunderstandings in the post-rebuttal response which we hope to clarify.
>
> **Q1) Difference Between NCI and NRI Training**
>
> We would like to clarify that the NRI models we compare with are also trained to predict 20 timesteps in futures (where during training time we supervise trajectory rollouts from NRI predictions for 20 timestep rollouts). Therefore, both NCI and NRI are trained using the same objective.
>
> **Q2) Inaccurate Notation in Equation 3**
>
> For the prediction experiments, the model encodes a partial trajectory x_partial into edge factors (represented as latent constraints in Eq 3). Then 21 timesteps are predicted, where the 1st is the fixed groundtruth that the model needs to have a point of reference in the prediction (as the baselines do).
> Equation 3 is **general** in the sense that it includes the option of reconstruction and prediction, where x_partial must be also sampled. However, in our forecasting experiments, **the EBM never sees the x_partial trajectory**. All information of x_partial is given by the encoded latent constraints.
> All timesteps of the prediction are sampled simultaneously with Langevin, except the 1st and excluding the trajectory observed by the encoder. Thus, the underlying expression in Equation 3 corresponds to what is implemented in our approach in the particular case of forecasting. We will clarify this in the final version of the paper.
>
> **Q3) Computational Constraints of Our Approach vs NRI**
>
> The computation of second order derivatives does not become intractable at a higher pace than NRI becomes intractable. NRI uses as a decoder 1 MLP/RNN cell per edge, applied recursively and with message passing. Therefore the burden increases quadratically with the number of nodes. In NCI, our decoder architecture is a set of two message passing graph networks across all nodes, and the computation of second order derivatives of this graph network with respect to input nodes also only grows quadratically with the number of nodes (as the computational cost is only additive across pairs of edges). We are happy to add explicit computational analysis showing these results if the reviewer desires.

---

### Official Review · Reviewer_ShW2 · 2022-10-24

**Confidence:** 3
**Correctness:** 2
**Technical Novelty And Significance:** 2
**Empirical Novelty And Significance:** 2
**Recommendation:** 3

**Clarity, Quality, Novelty And Reproducibility:**

I have few question about the paper:
* [Q1]: Does the encoder need a large number of training pairs? to capture the distinction between various Constrants?

* [Q2]: What effect do the Langevin Dynamic steps M have on the final outcome? How about the size of Constrants T?

`We illustrate how such a constraint decomposition of interactions enables more accurate long-horizon trajectory prediction performance over prior methods.`
`we illustrate how such a constraint decompositions of interactions is disentanglement, enabling the recombination of constraints between separate trajectories, as well as the addition of novel test-time constraints.`
* [Q3] I'm not quite grasping the concept; could you elaborate?

**Strength And Weaknesses:**

[Strength]
* S1: The research topic's inspiration is vivid and captivating.

[Weeknesses]
* W_1: The paper was somewhat challenging to follow, and the logical connection between several lines was unclear. The entirety of section 2 did not aid comprehension of literature.
* W_2: The empirical result is insignificant, and some baseline are absence(e.g. the recent work in section 2)

* W_other: some misused of cite and citep(Intro, para3)


**Summary Of The Paper:**

This paper presents an implicit method based on an energy model for discovering the underlying model in a dynamical system and forecasting agents' future dynamical processes. Instead of using a VAE-like solution, this paper proposed a EBM-based solution. On a variety of simulated data, the authors demonstrate the effectiveness of their algorithm and achieve superior results compared to a reasonable baseline.

**Summary Of The Review:**

Overall, I tend to vote for negative results and my main concern comes from innovation. I am not convinced that the proposed method is innovative enough, especially compared to previous work such as NRI. There is still room for further exploration of the proposed method in terms of its practicality, training efficiency, and robustness.

---

> ### Author Response · Authors · 2022-11-17
> **Comment for reviewer ShW2**
>
> We regret the reviewer concerns about clarity and novelty. We will modify our paper according to your comments and clarify your doubts as thoroughly as possible.
>
> > The paper was somewhat challenging to follow, and the logical connection between several lines was unclear. The entirety of section 2 did not aid comprehension of literature.
>
> We value the comments about clarity. Could the reviewer provide us with further examples? We will make modifications accordingly.
> **We modified Section 2 for clarity**. We will make further modifications if required.
>
> > The empirical result is insignificant, and some baseline are absence(e.g. the recent work in section 2).
>
> We included results from new baselines that have been suggested by reviewers in the general comment.
>
> > What effect do the Langevin Dynamic steps M have on the final outcome? How about the size of Constrants T?
>
> 1. We include the Langevin Dynamics steps ablation in the general comments.
> 2. If we understood correctly, there is an ablation regarding size of Constraints in Section A.5 of the Appendix.
>
> > Does the encoder need a large number of training pairs? to capture the distinction between various Constraints?
>
> Our encoder uses the same amount of training data as prior approaches. Thus, it does not need a large number of training pairs to learn the distinction between various constraints.
>
> > I'm not quite grasping the concept; could you elaborate?
>
> There are two typos in this paragraph: 1) disentanglement --> disentangled. 2) Addition of test-time constraints is not directly due to disentanglement as the paragraph suggests.
>
>  We appreciate the comment and modify this sentence in the main paper.
> As for the meaning, we are trying to summarize what has been said in the previous paragraph:
> 1. Our temporally one-shot prediction is better suited to long-horizon prediction, given that it prevents the error accumulation given by auto-regressive methods such as NRI. It is an important distinction from previous work that we energy functions are learned over **entire trajectories**. Our comparison to the Conditional-GNN baselines shows that one-shot prediction is only accurate if refined iteratively, as done by our constraint satisfaction optimization problem.
> 2. The fact that we can disentangle the edges allows for compositional generation in the following fashion: We generate the trajectory X that minimizes the energy for each edge latent factor **individually**. Therefore, at test time, we can combine edge factors from different samples and across datasets and generate a recombined trajectory.
> 3. Finally, as the energy minimization problem happens simultaneously for all energy functions, we can easily add a new hand-crafted energy function (e.g. a penalization for the velocity) and sample the trajectory X that satisfies this constraint as well as the set of learned energy functions.
>
> * [**Misuse of Citep**] We correct this error in the paper.

---

> > ### Author Response · Authors · 2022-12-08
> > **Further Clarifications?**
> >
> > Dear reviewer ShW2,
> >
> > We hope that we could answer your questions and improve our work in the desired directions. We made an effort to clarify our contributions with respect to the previous work and to clarify each one of your concerns. We also ran new experiments following your request. In case our response wasn't enough to clear your doubts, is there anything else we can clarify in the time we have left?

---

### Official Review · Reviewer_C8J2 · 2022-10-24

**Confidence:** 4
**Correctness:** 3
**Technical Novelty And Significance:** 3
**Empirical Novelty And Significance:** 3
**Recommendation:** 6

**Clarity, Quality, Novelty And Reproducibility:**

The paper is well written and the method is clearly explained. As discussed above, clarity of the qualitative results could be improved by providing videos. The appendix includes a detailed description of implementation details, which should suffice to reproduce the work. No code has been submitted, however.

The paper's proposed method appears novel, except for significant overlaps with a concurrent work (Rubanova et al., ICML 2022). The latter also proposes predicting trajectories by optimizing their adherence to a set of learned constraints. The main difference appears to be that the constraint function in Rubanova et al. is learned globally, whereas in the present work, their are inferred by an encoder network. This allows taking latent differences between interactions into account (e.g., charge polarity of particles). Additionally, the focus of the empirical evaluation is slightly different, with this paper putting a greater emphasis on disentanglement.


Questions and minor comments:
 - The attraction figure (Fig. 7) suggests that the particles' trajectories are printed starting with dark, opaque dots, which get increasingly lighter and more transparent as time passes. Intuitively, I would have expected the opposite direction. It might be useful to make this arrow of time more clear, at least in the few experiments in which it matters.
 - Is each predicted trajectory initialized entirely from noise, or is the known partial trajectory utilized?
 - How can the baseline models observe the randomized latent properties of the trajectories, such as the presence of springs or charge polarity? E.g., interaction networks by default only take a small number of prior observations into account, which might not be enough to infer interaction modes. Isn't this an unfair disadvantage?

**Strength And Weaknesses:**

Strengths:
 - The paper addresses an important problem in an interesting way, based on a clear motivation.
 - The method is sensible and clearly presented.
 - The quantitative empirical results are convincing and support the authors' claims.
 - The method's capabilities are thoroughly explored through additional experiments.

Weaknesses:
 - The novelty of the paper is put into question by a concurrent work (see below).
 - The qualitative results seem to generally match the claims in the text, but are somewhat hard to read from static images. It would be useful if the authors would provide video comparisons of the predicted trajectories.
 - The claim of long-term stability of the rollouts could be explored further, beyond the 20 timesteps measured in Table 1. While due to the chaotic nature of the systems considered, it cannot be expected that predicted trajectories exactly match ground truth beyond this point, long term stability may still be evaluated. For instance, by providing long video sequences, or by measuring the degree to which physical conservation laws hold (as in https://arxiv.org/pdf/1910.02425.pdf Fig. 5).




**Summary Of The Paper:**

The paper proposes neural constrain inference (NCI), a dynamics model which predicts future trajectories by inferring interactions between particles represented as sets of energy constraints, and sampling from the resulting energy-based model (EBM) using Langevin dynamics. It is demonstrated on various dynamics task that NCI yields lower rollout errors compared to methods based on standard feed-forward prediction, such as interaction networks. Additionally, various useful features of the method are demonstrated, namely disentanglement of interactions, the possibility of the recombination of different constraints, out-of-distribution prediction, and the introduction of novel hand-made constraints at test time.

**Summary Of The Review:**

Overall, this is a well written paper presenting an interesting idea, in a way that is well supported by experimental evaluations. While it could be argued that it offers limited novelty over Rubanova et al., I am of the opinion that when different aspects of a genuinely novel idea are concurrently developed by two papers, there is value in considering both perspectives. I am therefore leaning towards acceptance.

---

> ### Author Response · Authors · 2022-11-17
> **Comment for reviewer C8J2**
>
> We appreciate the positive and constructive comments from the reviewer. We will modify our paper according to your comments and address concerns below.
>
> > The novelty of the paper is put into question by a concurrent work.
>
> A deeper differenciation and comparisons from NCI to [1] can be found in general comments.
>
> > The qualitative results seem to generally match the claims in the text, but are somewhat hard to read from static images.
> > The attraction figure (Fig. 7) [...] It might be useful to make this arrow of time more clear, at least in the few experiments in which it matters.
>
> We attempt to answer both concerns, and provide an illustration of the NBA test-time constraints experiment in video. We attach a video of the two scenarios of the experiments where the arrow of time is clearer. The colors are flipped as suggested, as we agree with your assesment.
>
> > Is each predicted trajectory initialized entirely from noise, or is the known partial trajectory utilized?
>
> The states (both position and velocity) are initialized from uniform noise in [-1, 1]. For plotting, we use the velocity term of our state, and therefore the trajectories are plotted by accumulating velocity predictions.
>
> > How can the baseline models observe the randomized latent properties of the trajectories, such as the presence of springs or charge polarity?
>
> Indeed INs have no mechanism to infer interaction modes from the initial conditions. We included it as a baseline as previously requested by a reviewer, given its significance being the first relevant work on particle interaction networks. For other baselines, each of them has a different method to infer interactions. Based on message passing (Cond GNN, NRI or CBGN) or hidden state aggrupation (S-LSTM, TrajNet++ or LSTM joint). The remaining baselines (LSTM single, Static and IN) do not have a mechanism for it.
>
> > The claim of long-term stability of the rollouts could be explored further, beyond the 20 timesteps measured in Table 1. [...] Long term stability may still be evaluated.
>
> We provide an evaluation of the states energy as the squared distance among consecutive predicted steps, similarly to the provided reference. We evaluate ours and our main baseline (NRI) in the Charged dataset. Surprisingly, NRI shows a substantial decay while ours keeps a reasonable stability.
>
> [1] Yulia Rubanova and Alvaro Sanchez-Gonzalez and Tobias Pfaff and Peter W. Battaglia "Constraint-based graph network simulator" ICML 2022.

---

> > ### Author Response · Authors · 2022-12-08
> > **Further Clarifications?**
> >
> > Dear reviewer C8J2,
> >
> > We hope that we could answer your questions and improve our work in the desired directions. We made an effort to clarify our contributions with respect to the previous work and provided new experimental results that answer most of your concerns.
> >
> > Is there anything else we can clarify in the time we have left?

---

### Official Review · Reviewer_m9s9 · 2022-10-24

**Confidence:** 4
**Correctness:** 3
**Technical Novelty And Significance:** 2
**Empirical Novelty And Significance:** 2
**Recommendation:** 5

**Clarity, Quality, Novelty And Reproducibility:**

The paper is clearly written.
The quality of the experimental part is low, missing important comparisons.
The novelty is limited.
The work should be reproducible.

**Strength And Weaknesses:**

Strength:
- The paper is well-written and easy to follow
- Paper shows good empirical results.

Weaknesses:
- Limited novelty, the main contribution is replacing the VAE in NRI with an EBM.
- Unclear use of the term constraint. The so-called inferred constraints are simply the energy in the EMB. There is no real connection to any physical or mathematical constraint. I fail to see how this connects to constraints on the system, besides the fact that any generative model limits low-density areas.
- Missing comparisons in the experimental section. Tab. 2 does not include NRI results, and both tables do not show recent relevant baselines for comparison, e.g. Graber & Swhing "Dynamic Neural Relational Inference", Salzmann et al "Trajectron++: Dynamically-Feasible Trajectory Forecasting With Heterogeneous Data", and Rubanova et al "Constraint-based graph network simulator.".
- Regarding Rubanova et al, it is not "Concurrent to our work" as it has been on arxiv for almost a year now and published in ICML 2022.
- It is not clear how the "constraints" related to actual relations and the lackluster accuracy in prediction is not promising.


**Summary Of The Paper:**

The paper proposes a way to learn the dynamics between interacting particles by learning the constraints between them. It shows it can predict trajectories on synthetic and real datasets.

**Summary Of The Review:**

The two main issues I have with this paper are (1) It is unclear how the main concept of constraints is implemented besides the use of a EBM. (2)  The empirical evaluation is weak due to missing comparisons to more relevant and challenging baselines.

---

> ### Author Response · Authors · 2022-11-17
> **Comment for reviewer m9s9**
>
> We appreciate the positive and constructive comments from the reviewer. We will modify our paper according to your comments and address concerns below.
>
> > Unclear use of the term constraint. The so-called inferred constraints are simply the energy in the EMB. There is no real connection to any physical or mathematical constraint.
>
> We refer to edge EBMs as constraints for the following reason: To re-generate/predict trajectories, we find the trajectory X so that each one of the edge energies are satisfied (have the lowest value). However, **those act as "soft" constraints**, as they are not imposed to the solution. A similar yet different option would be using the word "potential", as the energy functions energy shape a potential field over trajectories. If the reviewer considers this term more adequate we will proceed to modify the current text to include it.
>
> > It is not clear how the "constraints" related to actual relations. Lackluster accuracy in prediction is not promising.
>
> The energy constraints are enforced to condition one and only one edge of the EBM. We use the FiLM modulation to modify the trajectory features according to the latent vector, at the edge stage of the message passing procedure. An illustration of the architecture can be found in Figure A.1. We can see how the latent features corresponding to the edge ij in the encoder condition only the features in edge ij of the EBM.
>
> > Missing comparisons in the experimental section.
>
> We added new baselines. See general comments.
>
> > Regarding Rubanova et al, it is not "Concurrent to our work"
>
> A deeper differenciation and comparisons from NCI to [1] can be found in general comments.
>
> > Limited novelty, the main contribution is replacing the VAE in NRI with an EBM
>
> Our submission leverages advances in Energy-Based models that bring new exciting properties to multi-agent trajectory modelling. This includes making non-trivial design choices that aren't immediately obvious. Following we describe key differences to NRI in each module of the architecture:
> 1. **Encoder**: we do use the same backbone  as NRI, but instead of a classification problem, we find continuous representations of edges that allow for a better tayloring of each specific scene. Especially in cases where relation-types are not a continuous set.
> 2. **Decoder**: There is a substantial number of differences with respect to NRI.
>     * **i)** **we predict temporally in one-shot**. This implies that we are learning energy functions over entire trajectories. We use the sampling process to refine our prediction instead of accumulating error with an auto-regressive approach.
>     * **ii)** our decoder's architecture is different. First, we are **decoding the whole trajectory with a single graph network**, while NRI either recurrently employs a single MLP or an LSTM cell to predict states. Second, **we look at the energy from a local to a global perspective with two branches** (see architectural details in Sect. A.2.).
>     * **iii)** **the decoding process is a GNN forward pass, reversed**. Given that the updates follow gradient-based langevin steps. This is equivalent to employing deconvolution instead of convolution for CNNs.
>     * **iv)** **Our edge-conditioning by FiLM formulation is fundamentally different from the gating function implemented by NRI**, and the optimization process allows for recombination, OOD detection and test-time constraint addition that cannot be done by common feed-forward networks such as in NRI.
>
> [1] Yulia Rubanova and Alvaro Sanchez-Gonzalez and Tobias Pfaff and Peter W. Battaglia "Constraint-based graph network simulator" ICML 2022.

---

> > ### Comment · Reviewer_m9s9 · 2022-11-25
> > **Thank you**
> >
> > Thank you for your changes, I do think it improves the paper, but finally, I still think the novelty is limited and the results aren't convincing enough so I keep the score at 5.

---

> > > ### Author Response · Authors · 2022-12-08
> > > **Further Clarifications?**
> > >
> > > Dear reviewer m9s9,
> > >
> > > We regret to hear that you do not agree with our claims after our rebuttal. If there is a particular point that is not convincing to you, please let us know and we will try to clarify it further. In any case, we appreciate the time you took for reviewing our work and answering to our rebuttal.

---

> > > > ### Comment · Reviewer_m9s9 · 2022-12-12
> > > > **No need for further clarifications**
> > > >
> > > > Dear authors,
> > > > In the end, I gave a score of marginally below the acceptance threshold so I do think the paper has merit. However, my concerns are with the novelty and significance which are subjective measures.

---

### Author Response · Authors · 2022-11-17
**General Comments 1/2**

We thank reviewers for their constructive comments and feedback. We are glad that reviewers generally found that our work is technically reasonable and complete (1k6S), well written and presented (m9s9, C8J2, 1k6S), well motivated (C8J2), new and original (1k6S), with thorough and convincing empirical results (m9s9, C8J2) and supported claims (C8J2). However, some reviewers had concerns about the novelty and underlying evaluation of our approach.
## 1 Novelty
### 1.1 Novelty with respect to [1].
Concerns were raised when it comes to comparing NCI to the recent work in [1]. Our work has been previously submitted to NeurIPS 2022, where we claimed concurrency with [1], as the paper came out in ICML when NCI was being completed. We understand that concurrency could be now under doubt, and therefore we remove it from our paper.
[1] However, we believe that our work is fundamentally different from [1] in the following aspects (also highlighted by C8J2).
1. In NCI, **energy functions are learned over entire trajectory states**. In contrast [1] directly learns an energy only to predict the next state of a trajectory. This settings enables long-term predictions naturally accumulate less error, as we do not need to fix a state value before predicting the next. This also substantially reduces the number of optimization steps needed to generate a trajectory, as optimization is done simultaneously for every time-step. This is also fundamentally different from other baselines such as NRI. Finally, note that by defining energy functions over an entire trajectory -- constraints that effect behavior across many timesteps may be more naturally recovered.
2. **Our model learns a set of different energy functions per edge**, as opposed to a single one. In [1] a global energy function is learned for all nodes, by leveraging ground truth attributes such as mass, and assuming that dynamics and interactions among objects only vary with respect to those attributes. Thus [1] has no mechanism to predict trajectories in the abscence of those attributes nor when different types of relations are present among objects. In contrast, our approach directly discovers such energy functions without any explicit annotations.
3. **Learning different energy functions per edge is not an easy task**. We must balance reasonable computation with disentanglement of the edges. Having one EBM per edge would result in a huge computational burden. Hence we propose to use a GNN-based EBM, where every edge ij is targeted by its corresponding edge latent vector ij by means of the FiLM modulation. For an illustration of this process see the architecture figure in Fig. A1 of the appendix.
Beyond targeting edges, we train two instances of our EBM, which are employed simultaneously to generate the predicted trajectories. Each EBM will handle a random subset of the edges by masking out the rest. In consequence, there is no communication among the two subsets in the message-passing process. However, both gradients are used to generate the new trajectories.
3. **The resulting energy functions are decoupled.** This allows NCI to infer the correct interactions but also to **recombine edge constraints from different trajectories**, by simultanously optimizing different sets of energy functions. Such an ability is not available for [1]

Code for [1] has not been made public. Hence, we run our implementation of [1], where a single EBM has access to the full observed history and optimizes over 1 future timestep. In the following table, we present results in the Charged dataset for 1 step rollout predictions. Results show a fairly similar prediction MSE in step 1, but a significant degradation as time unrolls.
| Charged Dataset (MSE)   | 1           | 10          | 20          |
|-------------------------|-------------|-------------|-------------|
| CBGN                    | 1.14e-3     | 3.35e-2     | 1.06e-1     |
| NCI                     | **8.85e-4** | **3.33e-3** | **6.54e-3** |
### 1.2 Novelty in terms of results
Reviewers had several concerns on our quantitative results on prediction error. While we preserve better long-term performance than the baselines we would like to highlight that our understanding of NCI's contribution is not centered in prediction performance.
Quantitative results are meant to show that NCI preserves a good prediction power, in pair with methods that are specifically designed for prediction. While this shows that NCI can generate energy functions around trajectories that are accurate, we present new exciting qualities that naturally emerge from our approach. This includes trajectory mixing, OOD detection, addition of constraints in test-time or tools for interpretability of our model.

[1] Yulia Rubanova and Alvaro Sanchez-Gonzalez and Tobias Pfaff and Peter W. Battaglia "Constraint-based graph network simulator", ICML 2022.

---

> ### Author Response · Authors · 2022-11-17
> **General Comments 2/2**
>
> ## 2. Additional Results
> ### 2.1 Ablation Study
> We test some of our model capabilities suggested by the reviewers. In particular varying the following parameters: (ShW2) Number of langevin dynamics steps and (1k6S) Number of nodes in train vs test.
>
> | Charged Dataset (MSE)   | 1           | 10          | 20          |
> |-------------------------|-------------|-------------|-------------|
> | **Num. Langevin Steps** |             |             |             |
> | M=1                     | 1.06e-1     | 1.18e-1     | 2.47e-1     |
> | M=2                     | 3.23e-2     | 4.34e-2     | 8.22e-2     |
> | M=3                     | 7.00e-3     | 1.16e-2     | 2.68e-2     |
> | M=4                     | 1.86e-3     | 4.63e-3     | 1.00e-2     |
> | M=5                     | 1.03e-3     | 3.51e-3     | 7.05e-3     |
> | M=6 (Train)             | **8.85e-4** | **3.33e-3** | 6.54e-3     |
> | M=7                     | 9.00e-4     | **3.33e-3** | **6.45e-3** |
> | M=8                     | 8.93e-4     | 3.36e-3     | 6.47e-3     |
> | M=9                     | 9.00e-4     | 3.39e-3     | 6.60e-3     |
> | M=10                    | 9.10e-4     | 3.44e-3     | 6.66e-3     |
> | M=20                    | 1.03e-3     | 3.64e-3     | 7.04e-3     |
> | **Num. Nodes**          |             |             |             |
> | N=3                     | 1.79e-3     | 3.71e-3     | 6.74e-3     |
> | N=5 (Train)             | **8.85e-4** | **3.33e-3** | **6.54e-3** |
> | N=7                     | 1.34e-3     | 4.66e-3     | 1.01e-2     |
>
> ### 2.2. New Baselines
> Following our reviewers requests, we ran experiments in the realistic datasets (NBA, JPL Horizons) with NRI and dNRI [2] baselines. For NRI we use the dynamic decoder, which is shown to have better results in realistic datasets such as NBA.
> For dNRI, we do our best effort with the authors' code, but performance is inferior than expected:
> | NBA dataset (MSE) | 1       | 10      | 20      |
> |-------------------|---------|---------|---------|
> | NRI               | **3.56e-6** | 7.46e-4 | 2.74e-3 |
> | dNRI [2]          | 7.97e-6 | 1.07e-3 | 4.52e-3 |
> | NCI               | 1.27e-5 | **3.46e-4** | **1.86e-3** |
>
> | NBA dataset (MSE) | 1           | 10          | 20          |
> |-------------------|-------------|-------------|-------------|
> | NRI               | **2.67e-7** | 7.35e-7     | 1.16e-6     |
> | dNRI [2]          | 1.35e-5     | 5.12e-5     | 1.64e-4     |
> | NCI               | 4.05e-7 | **4.70e-7** | **8.60e-7** |
> ### 2.3 Longer-time rollouts
> As suggested by reviewers C8J2 and 1k6S we evaluate our model and NRI in longer term prediction. Given the chaotic nature of trajectories, we evaluate the energy among consecutive states (squared pairwise difference) as suggested by C8J2. R1 indicates that we iteratively predict 1 step 100 times. We evaluate it for the Charged dataset, which is our most challenging synthetic dataset. Results show how NCI preserves approximately more than 3 times of the energy than NRI as time unfolds.
> | Charged Preds Energy (1e-3) | 1-2 | 10-11 | 20-21 | 30-31 | 40-41 | 50-51 | 60-61 | 70-71 | 80-81 | 90-91 | 100-101 |
> |-----------------------------|-----|-------|-------|-------|-------|-------|-------|-------|-------|-------|---------|
> | Baseline                        | 5.0 | 4.9   | 5.0   | 5.0   | 4.9  | 4.9   | 5.0  | 4.9   | 5.0  | 5.0   | 4.9     |
> | NCI  (R1)   | 3.2 | 2.4   | 2.2   | 2.2   | 2.1   | 2.1   | 2.0   | 1.9   | 1.9   | 2.0   | 2.0     |
> | NRI                   | 1.2 | 0.75  | 0.65  | 0.60  | 0.68  | 0.62  | 0.61  | 0.70  | 0.59  | 0.57  | 0.65    |
>
> ### 2.4 Additional Visualizations
> We included videos of the results in additional constraints qualitative experiments for NBA dataset.
>
> ## 3 Minor revisions
> Some reviewers pointed out the misuse of the citation format. We corrected it in the paper.
>
> [1] Yulia Rubanova and Alvaro Sanchez-Gonzalez and Tobias Pfaff and Peter W. Battaglia "Constraint-based graph network simulator", ICML 2022.
>
> [2] Colin Graber and Alexander G. Schwing, "Dynamic Neural Relational Inference", CVPR 2020.

---

### Decision · Program_Chairs · 2023-01-20

**Decision:**

Reject

**Justification For Why Not Higher Score:**

Overall, this paper makes an interesting contribution on top of NRI [1] and – while keeping roughly comparable performance to NRI on existing tasks – adds new capabilities such as OOD detection. This paper however only scratches the surface of investigating these new capabilities (in a mostly qualitative nature) and the comparison to NRI and other NRI-related methods does not show clear advantages over existing work. While this paper could be considered for acceptance, I agree with the reviewers that it does not quite meet the bar in terms of significance in its current form.

**Justification For Why Not Lower Score:**

N/A

**Metareview: Summary, Strengths And Weaknesses:**

This paper investigates modeling interacting dynamical systems using a constraint-based energy-based model (EBM). The proposed method, neural constraint inference (NCI), discovers relational constraints by reconstructing multi-object/multi-particle input sequences using an EBM-based approach as opposed to a VAE-based approach used in prior work (NRI [1]). The method is validated in trajectory prediction and relation/interaction discovery and shows roughly comparable performance when evaluated in the same setting as NRI [1] (better sequence reconstruction, decreased interaction recovery performance).

The reviewers highlighted that the paper is well-written, addresses an important problem, and makes interesting contributions that enable OOD detection and inclusion of user-defined constraints, which NRI is not capable of.

All reviewers, however, share concerns around novelty of the approach (compared to [1,2]) and the significance of the demonstrated results. Claims around novel capabilities of the model (OOD detection, inclusion of soft constraints at test time) are primarily validated qualitatively or in very limited synthetic settings.

In summary, I agree with the concerns of the reviewers that this paper does not quite meet the bar for acceptance in terms of significance of the results and in terms of novelty of the method and overall setting compared to prior works [1,2].

For a future revision, I would recommend that the authors further investigate and evaluate the novel qualities enabled by their model, i.e. OOD detection, inclusion of constraints, and demonstrate them on a challenging task, while reducing the focus on the direct comparison to NRI in terms of trajectory prediction / relation recovery, where the benefits are less clear.

During the (closed) discussion period, the reviewers have further highlighted the following suggestions/concerns:
* The new dNRI baseline is insightful, but fails to replicate performance gains over NRI and might therefore not be fully convincing
* The new energy conservation experiments for long-term rollouts partially help address concerns around long-term stability, but there is still a substantial deviation from the ground truth
* It would be insightful to train NCI with fewer time steps (e.g. 1 if possible, or only a small number) and perform an explicit roll-out for 10 or 20 time-steps to compare both NRI and NCI in an auto-regressive manner

[1] Kipf et al., Neural Relational Inference for Interacting Systems (ICML 2018)
[2] Rubanova et al., Constraint-based graph network simulator (ICML 2022)